# Chromatin landscape at *cis*-regulatory elements orchestrates cell fate decisions in early embryogenesis

Francesco Cardamone [1,2,3,12], Annamaria Piva [4,12], Eva Löser [1], Bastian Eichenberger[4], Mari Carmen Romero-Mulero [1,2], Fides Zenk[5], Emily J. Shields[6,7,8], Nina Cabezas-Wallscheid [1,9,10], Roberto Bonasio [6,7], Guido Tiana[11], Yinxiu Zhan [4] ✉ & Nicola Iovino [1] ✉

The establishment of germ layers during early development is crucial for body formation. The *Drosophila* zygote serves as a model for investigating these transitions in relation to the chromatin landscape. However, the cellular heterogeneity of the blastoderm embryo poses a challenge for gaining mechanistic insights. Using 10× Multiome, we simultaneously analyzed the in vivo epigenomic and transcriptomic states of wild-type, E(z)-, and CBP-depleted embryos during zygotic genome activation at single-cell resolution. We found that pre-zygotic H3K27me3 safeguards tissue-specific gene expression by modulating *cis*-regulatory elements. Furthermore, we demonstrate that CBP is essential for cell fate specification functioning as a transcriptional activator by stabilizing transcriptional factors binding at key developmental genes. Surprisingly, while CBP depletion leads to transcriptional arrest, chromatin accessibility continues to progress independently through the retention of stalled RNA Polymerase II. Our study reveals fundamental principles of chromatin-mediated gene regulation essential for establishing and maintaining cellular identities during early embryogenesis.

The transition of a pluripotent zygote into a metazoan with specialized cell types occurs with remarkable speed and precision. The *Drosophila melanogaster* embryo continues to serve as a valuable model for uncovering how cellular identities are established. After fertilization, cycles of nuclear divisions generate a syncytial embryo with approximately 6000 nuclei that migrate toward the periphery and become surrounded by cell membranes at cycle 14[1,2]. Transplantation experiments and in situ hybridization experiments suggest that cellular identities are already primed at cycle 14 (stage 5)[1,3–6]. At this stage, the zygotic genome is activated, and cells exhibit spatial gene expression and chromatin accessibility patterns that respond to positional cues along the anteroposterior and dorsoventral axes[7–13]. ZGA is facilitated by transcription factors, including pioneers such as Zelda and GAGA factor[14–22], as well as histone modifications and histone variants[23–27].

[1]Max Planck Institute of Immunobiology and Epigenetics, Freiburg, Germany. [2]Faculty of Biology, University of Freiburg, Freiburg, Germany. [3]International Max Planck Research School of Immunobiology, Epigenetics and Metabolism (IMPRS-IEM), Freiburg, Germany. [4]Department of Experimental Oncology, European Institute of Oncology, IRCCS, Milan, Italy. [5]Epigenomics of Neurodevelopment, Brain Mind Institute, School of Life Sciences, EPFL – Ecole Polytechnique Federal Lusanne, Ecublens, Switzerland. [6]Epigenetics Institute, Department of Cell and Developmental Biology, University of Pennsylvania Perelman School of Medicine, Philadelphia, PA, USA. [7]Department of Cell and Developmental Biology, University of Pennsylvania Perelman School of Medicine, Philadelphia, PA, USA. [8]Department of Urology and Institute of Neuropathology, Medical Center–University of Freiburg, Freiburg, Germany. [9]Laboratory of Stem Cell Biology and Ageing, Department of Health Sciences and Technology, Swiss Federal Institute of Technology (ETH Zürich), Zürich, Switzerland. [10]Centre for Integrative Biological Signalling Studies (CIBSS), Freiburg, Germany. [11]Università degli Studi di Milano and INFN, Milan, Italy. [12]These authors contributed equally: Francesco Cardamone, Annamaria Piva. ✉e-mail: yinxiu.zhan@ieo.it; iovino@ie-freiburg.mpg.de

Spatiotemporal regulation of the epigenome has been revealed to shape the distinct transcriptomic networks that ensure proper development[23,24,26–35]. Cell type-specific transcription is frequently regulated by enhancers, whose activities, in turn, are activated and repressed by acetylation and methylation of histone H3 lysine 27 (H3K27), respectively[27,36–38]. These histone modifications are deposited by the acetyltransferase CBP/p300 complex and the methyltransferase Polycomb repressive complex 2 (PRC2)[24,27,28,30–32,35,39–55]. Established *cis*-regulatory networks confer unique transcriptional patterns to each cell type and have been shown to play a pivotal role at later stages of development[7,56–65]. Nevertheless, the extent to which the epigenetic landscape contributes to the formation of germ layer identities in the early ZGA *Drosophila* embryo, before gastrulation, remains unclear.

Here, we explore how chromatin dynamics during ZGA regulate germ layers precursors establishment. By using multiple chromatin profiling and transcriptomic approaches at both bulk and single-cell resolution, we identify cell type-specific enhancer accessibility as a key determinant of gene expression. Furthermore, we demonstrate that *cis*-regulatory elements activity is ensured by precise epigenetic regulation mediated by E(z) and CBP, along with their respective histone modifications H3K27 trimethyl and acetyl. This mechanism ensures robust control of cell fate, preventing the emergence of undifferentiated states at the onset of zygotic transcription and facilitating proper embryogenesis.

## Results

### H3K27me3 and H3K27ac are distributed differently across germ layers in the early embryo

To assess the relationship between the epigenome and transcriptome during early embryo development, we precisely collected wild-type *Drosophila* embryos in three stages of embryogenesis that span the first 2.5 h after fertilization: (i) before cycle 9, when transcription has not been initiated and the nuclei are still totipotent, (ii) cycle 10–12, when some nuclei have moved to the periphery, and the minor wave of ZGA initiates (around 600 genes transcribed) and (iii) cycle 14, when cellularization occurs, and the major wave of ZGA begins (6000 genes transcribed) (Fig. 1a and Supplementary Fig. 1a)[23,66].

We used immunofluorescence to detect the opposing and mutually exclusive modifications of H3K27: H3K27me3 (a marker of repressed transcription) and H3K27ac (a marker of active transcription). We found that both were present at all three developmental stages, aligning with previous findings (Fig. 1b and Supplementary Fig. 1b)[24,27]. In particular, H3K27me3 is inherited from the maternal germline[27,30], while H3K27ac is established de novo after fertilization[24,27]. Before cycle 9, H3K27me3 formed distinct patterns that were non-overlapping with regions marked by H3K27ac, which were more uniformly distributed along the chromosome arms and in pericentromeric regions (Fig. 1b and Supplementary Fig. 1b). This pattern was evident in lateral sections of cellularized embryos (Supplementary Fig. 1a, b).

To study the distribution of these marks on chromatin, we performed CUT&Tag as well as ATAC-seq across early embryogenesis. To compare different stages, we adopted total histone 3 (H3) CUT&Tag as a control to normalize for the number of nuclei at each developmental time point. Additionally, we incorporated a spike-in to ensure quantitative accuracy in our measurements (Supplementary Fig. 1c). As development progressed, the enrichment of H3K27me3 and H3K27ac on chromatin increased, as did chromatin accessibility (Supplementary Fig. 1d–g)[67]. Intriguingly, our CUT&Tag revealed putative ambivalent regions across all three stages, bearing both H3K27me3 and H3K27ac (Fig. 1c and Supplementary Data 1), as illustrated by the *toy* and *opa* loci (Fig. 1d and Supplementary Fig. 1h). Presumed ambivalent peaks and total peaks of H3K27me3 and H3K27ac were similarly distributed throughout the genome (Supplementary Fig. 2a). Additionally, we could identify in cycle 14 embryos, ZGA-specific

ambivalent regions (Fig. 1e and Supplementary Data 1). Genes associated with these peaks are involved in developmental processes, cell fate commitment, and embryonic morphogenesis (Supplementary Fig. 2b).

Bivalent chromatin domains enriched for both H3K4me3 and H3K27me3 at poised genes have been very well documented in vertebrates[55,68–73]. However, in *Drosophila*, only H3K4me1 and H3K27me3 bivalency has been described[44,74]. Additionally, no evidence supports the presence of regions bearing both H3K27me3 and H3K27ac[44,75]. Therefore, we speculated that these putative ambivalent regions of H3K27me3 and H3K27ac arose from the differential epigenetic status across distinct cell types within the whole embryo, leading to the assumption that a possible cell type-specific distribution of these two marks could be already in place at these early developmental stages. Using a known germ layer marker for mesoderm, we FACS-sorted mesoderm nuclei from embryos during gastrulation and performed ChIP-seq (Supplementary Fig. 2c)[44]. We compared the chromatin landscape of this mesoderm cell subset to the CUT&Tag data from the whole cycle 14 embryos and found that sorted mesoderm nuclei did not exhibit any ambivalence. (Fig. 1e and Supplementary Fig. 2d). Furthermore, the repressive H3K27me3 mark was significantly enriched within genes that are not expressed in the mesoderm (Supplementary Fig. 2e)[76]. These results suggest that a distinct, mutually exclusive distribution of these two histone modifications may be already defining different cell subsets as early as ZGA.

### Enhancer opening is associated with cell fate commitment at ZGA

To address whether this mechanism is present in all embryonic cells during ZGA, we explored the epigenome and transcriptome landscape of the whole embryo at the single-nucleus level. Briefly, we isolated nuclei from hand-sorted wild-type embryos at cycle 14 (ZGA) and performed scATAC-seq and scRNA-seq of two biological replicates using 10× Multiome (Fig. 2a). Clustering and uniform manifold approximation and projection (UMAP) based on gene expression, genome-wide chromatin accessibility, and their integration, revealed well-defined cell clusters (Fig. 2b and Supplementary Fig. 3a, b). Annotation of the principal germ layers (mesoderm, ectoderm, and endoderm) and their sub-types (e.g., anterior-posterior ectoderm, ventral ectoderm, anterior endoderm) was based on the BDGP in situ database (insitu.fruitfly.org) and previously published marker genes[76] (Supplementary Data 2). Transcript levels and genomic accessibility of marker genes correlated in all germ layers (Supplementary Fig. 3c). Thus, the progenitors of the different embryonic cell types are already well established at this early developmental stage, not only at the transcriptomic level but also at the chromatin accessibility level. We classified an additional cluster of nuclei as yolk cells, which provide metabolic support for the developing embryo[76–79], and a small cluster that we named "undifferentiated cells" exhibiting no expression of germ layer-specific marker genes (Fig. 2b and Supplementary Fig. 3a and c).

Next, we examined whether the accessibility of specific *cis*-regulatory elements, such as enhancers and promoters, define cell identity at ZGA. To this end, we focused on highly transcriptomically variable genes (HVGs) in our dataset and performed UMAP visualization based on (i) the most accessible promoter of those genes or (ii) the enhancer with the highest correlation to gene expression (see Methods and Supplementary Fig. 3d). Intriguingly, enhancer accessibility could define the different germ layers resembling the transcriptomic embedding, whereas promoters did not (Fig. 2c and Supplementary Data 3). Among those putative enhancers, 62% overlapped with a curated list of regulatory regions[80], underscoring the reliability of our method (Supplementary Data 3). Additionally, gene expression correlated with the accessibility of enhancers rather than promoters (Fig. 2d), in line with previous findings at later developmental stages (4

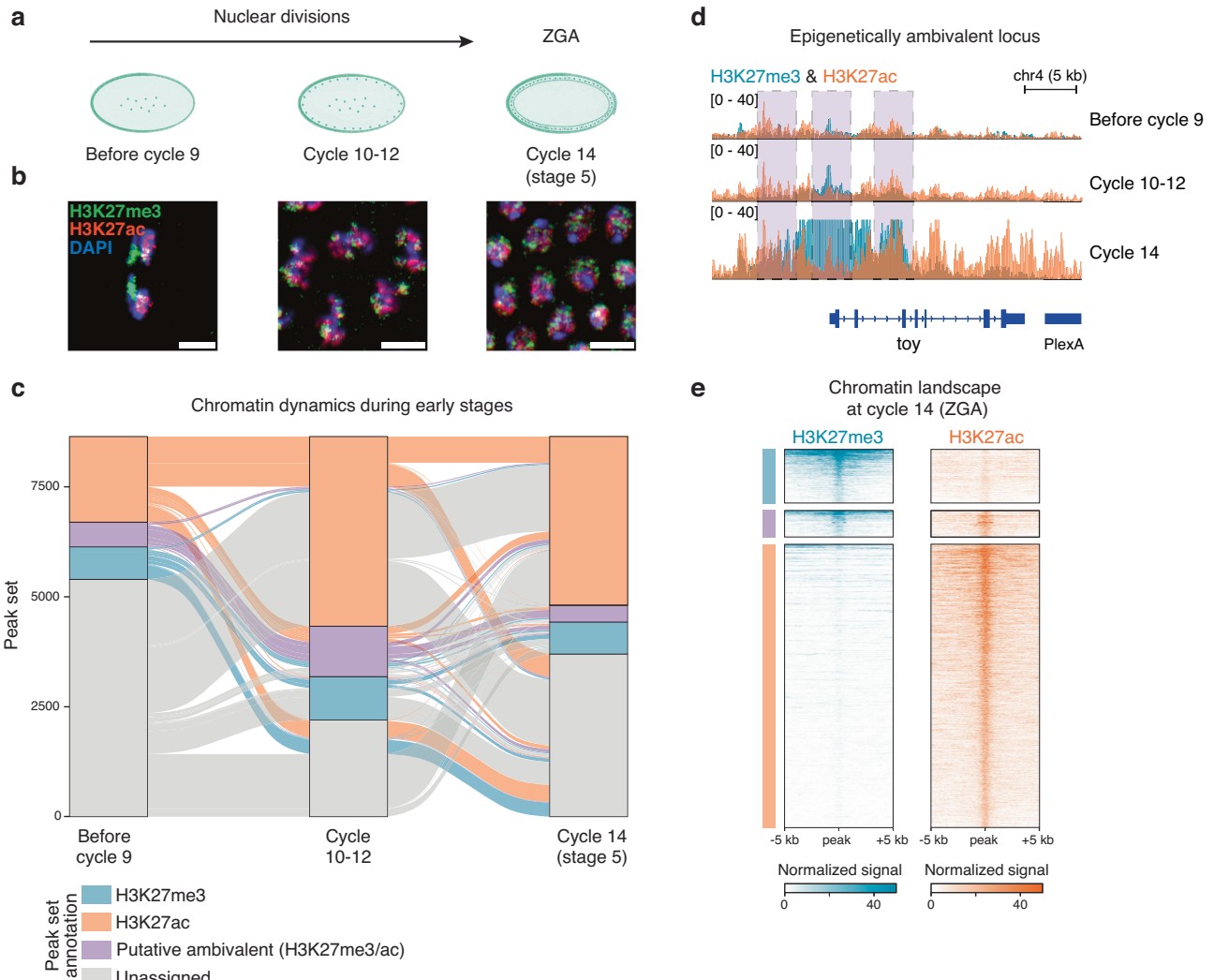

**Fig. 1 | Histone H3K27 trimethylation and acetylation dynamics during early embryonic development. a** Schematic representation of early embryonic development. After fertilization, the *Drosophila* embryo undergoes rapid and synchronous cycles of nuclear divisions. By cycle 14 (stage 5), nuclei migrate to the periphery of the syncytium, get cellularized and transcriptionally activate their genome (ZGA). **b** Representative immunofluorescence staining from three biological replicates across early embryonic development. H3K27me3 and H3K27ac localize to chromatin before ZGA. Scale bar, 5 μm. **c** Chromatin dynamics during early embryonic development. The alluvial plot shows the epigenetic states from before cycle 9 to cycle 14, based on H3K27me3, H3K27ac, H3K27me3/ac (putative ambivalent) or unassigned peaks. Blue cluster represents H3K27me3 specific peaks, purple cluster represents putative ambivalent peaks, and orange cluster represents H3K27ac specific peaks. Gray clusters represent unassigned peaks in each time point. **d** Genome browser snapshot of a putative ambivalent locus throughout early embryogenesis from bulk CUT&Tag normalized profiles (see Methods) for H3K27ac (in orange) and H3K27me3 (in blue). The putative ambivalent peaks are highlighted by purple dashed boxes. **e** Heat maps of normalized CUT&Tag signal (see Methods) for H3K27me3 and H3K27ac at cycle 14 (ZGA). The intensity of the signal is centred on peak cathegories specific to cycle 14, ranked based on H3K27me3. Blue cluster represents H3K27me3 peaks, purple cluster represents ambivalent peaks (H3K27me3/ac) and orange cluster represents H3K27ac peaks.

to 6 h embryos[59,60]. The same was observed for marker genes, suggesting that their expression is instructed preferentially by enhancers (Supplementary Fig. 3e). To support our findings with an orthogonal approach, we performed clustering and visualization in our scATAC-seq embedding based on previously annotated enhancer peaks[80], confirming that enhancers are better definers of cell types already at the ZGA stage (Fig. 2c and Supplementary Fig. 3f). To exemplify this at single-cell level, we looked at gene expression and chromatin accessibility for germ layer specific marker genes, such as *Mes2* (mesoderm marker), *Lim2* (anterior endoderm marker), and *fkh* (posterior endoderm marker). Interestingly, enhancer accessibility correlated with the expression of these genes in the same nuclei, whereas promoters were ubiquitously open regardless of gene expression (Fig. 2e–h and Supplementary Fig. 3g, h). Our single-cell findings expand on the ChIP-seq results of H3K27me3 and H3K27ac in nuclei from mesoderm cells,

suggesting that the mutually exclusive enrichment of these two marks is likely not limited to mesoderm but is also a feature of ZGA embryos, as inferred from the differential chromatin accessibility measured by scATAC-seq. This implies that the specification of different chromatin states across different germ layers starts quite early in the embryo, already at the ZGA stage (Fig. 2g, h and Supplementary Fig. 3h). On the other hand, RNA polymerase II (RNAPII) was present both at mesoderm and other germ layer specific genes in mesoderm sorted nuclei, in line with previous findings, suggesting that promoter accessibility is maintained across different cell types by a possibly paused RNA polymerase independently of cellular identity, while productive transcription is governed by tissue specific enhancers (Fig. 2g, h and Supplementary Fig. 3h)[63,81–83].

Moreover, to understand the timing of germ layer specification during embryogenesis, we generated bulk RNA-sequencing

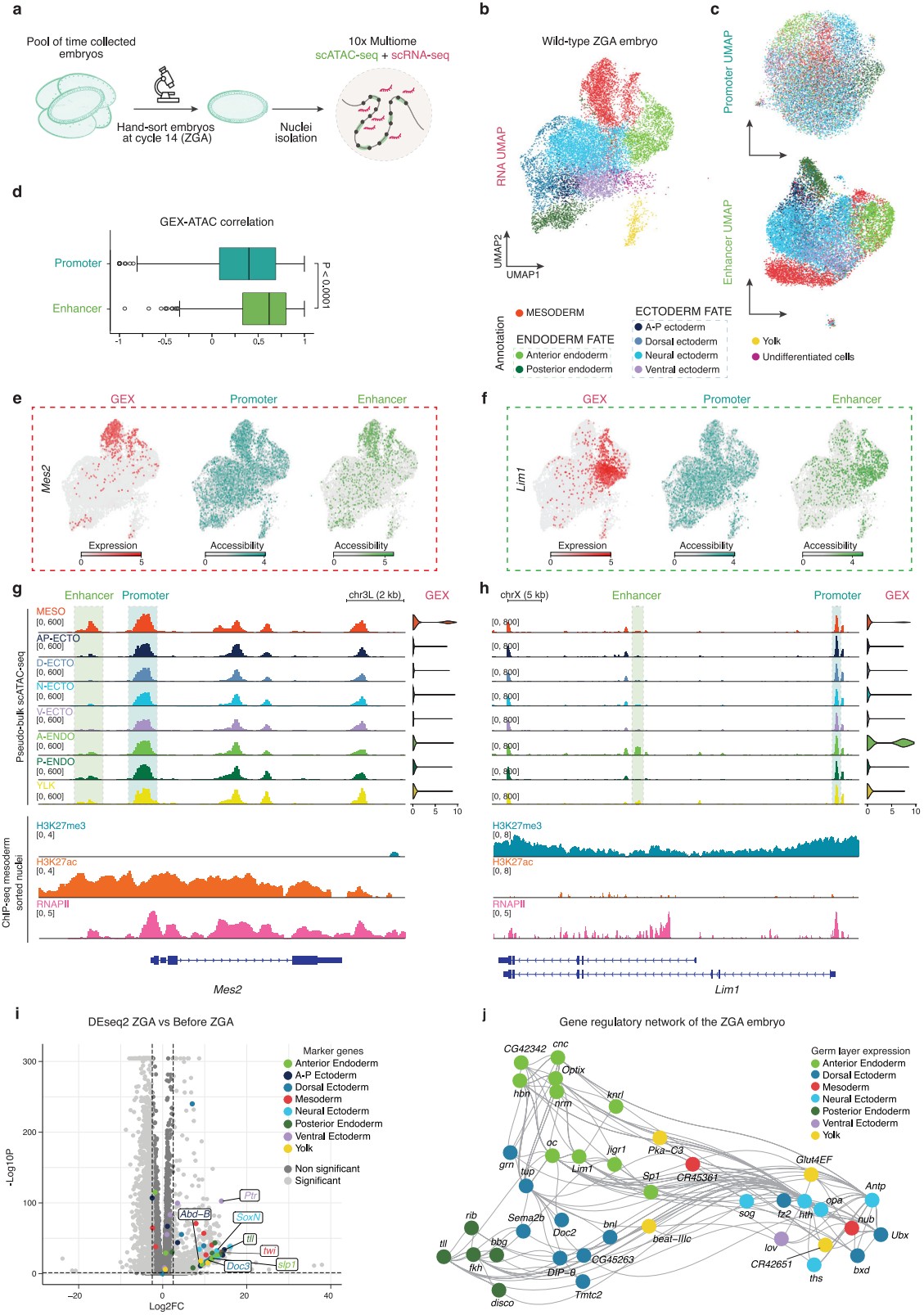

(RNA-seq) from (i) early quiescent embryos (before cycle 9) and (ii) cycle 14 embryos during ZGA. We examined the expression levels of our set of marker genes used for annotating the multio-mic dataset (Supplementary Fig. 3c and Supplementary Data 2). Differential expression analysis showed that cell identity markers are actively expressed at the time of ZGA but not earlier, sug-gesting that, while positional informations are instructed by maternally deposited factors[8], germ layer specification occurs exclusively with the awakening of the zygotic genome (Fig. 2i). Consequently, gene regulatory network (GRN) analysis[84] based on our ZGA scRNA-seq and scATAC-seq revealed the cooperative regulation of cell type specification by key regulators as *tll* and *sog*, which were expressed in posterior endoderm and neural ecto-derm, respectively (Fig. 2j).

**Fig. 2 | Single-cell multiomic profiling reveals enhancer-guided early cell fate commitment in ZGA embryos. a** Experimental design and schematic representation of collection strategy and manual selections of ZGA embryos. Nuclei are isolated and processed for 10× Multiome (scATAC-seq + scRNA-seq). Adapted from an image created in BioRender. Cardamone, F. (2025) https://BioRender.com/a06c492. **b** UMAP embedding of scRNA-seq data of two integrated biological replicates of the ZGA embryo. Cluster identities were assigned by expression of marker genes. A-P ectoderm, anterior-posterior ectoderm. **c** UMAP embedding of scATAC-seq based on either promoter peaks or enhancer peaks associated to the highly variable genes, highlighting the contribution of enhancer regions in driving germ layers definition. **d** Distribution of spearman correlation between gene expression (GEX) and accessibility (ATAC) of highly variable genes (HGVs) at their most accessible promoter peak or at their highest score linked peak (enhancer), in each single-cell. Boxes center refers to mean, lower and upper quartiles (Q1 and Q3, respectively). Whiskers, 1.5 × IQR below Q1 and above Q3. Outliers are shown. Two-sided Mann–Whitney $U$ test. $P = 3.45 \times 10^{-22}$. **e, f** Gene expression (GEX) or chromatin accessibility of most accessible promoter peak or highest score linked peak

(enhancer) of *Mes2* (mesoderm marker gene) and *Lim1* (anterior endoderm marker gene). Dashed boxes highlight the germ layer where the gene is expressed. **g, h** Genome browser snapshot of *Mes2* and *Lim1* loci. Top, aggregated ATAC reads of each germ layer and violin plot of the respective *Mes2* or *Lim1* gene expression (GEX). Promoter is highlighted by blue dashed box, enhancer is highlighted by green dashed box. Bottom, ChIP-seq signal of H3K27me3 (this study), H3K27ac and RNA polymerase II (RNAPII) from mesoderm sorted nuclei[44]. MESO, mesoderm. AP-ECTO, anterior-posterior ectoderm. D-ECTO, dorsal ectoderm. N-ECTO, neural ectoderm. V-ECTO, ventral ectoderm. A-ENDO, anterior endoderm. P-ENDO, posterior endoderm. YLK, yolk. **i** Volcano plot showing differential gene expression analysis in ZGA versus before ZGA embryos. Dot colour represents the marker genes used to annotate the 10× Multiome dataset or significance. Two-sided Wald test. Source data are provided as Source Data file. **j** Gene regulatory network of the wild-type ZGA embryo inferred by Pando[84] on the 10× Multiome dataset. Dot colour represents the annotated germ layer of the cell with the highest gene expression value for each gene.

Overall, our data suggest that ZGA represents the earliest time point in embryogenesis in which cell type specific enhancers become active and initiate the transcriptional programs essential for germ layer specification.

## E(z) and CBP depleted embryos display aberrant ZGA and failure in completing embryogenesis

E(z) and CBP are responsible for H3K27me3 and H3K27ac modifications, respectively[24,27,85–87]. To explore the roles of these enzymes on the establishment of cell type-specific transcriptomes and chromatin states at ZGA, we generated embryos with maternal depletion of each factor at both RNA and protein levels (Supplementary Fig. 4a)[24,27].

By immunofluorescence, we observed a dramatic loss of H3K27me3 and H3K27ac in embryos depleted of E(z) and CBP, respectively (Supplementary Fig. 4b). Additionally, DAPI staining of cycle 14 embryos did not highlight dramatic differences in morphology nor nuclei number across conditions, except for the expected twisted phenotype and a slight reduction of nuclei upon CBP depletion[88] (Supplementary Fig. 4c). However, only about 20–30% of the CBP-KD and E(z)-KD embryos reached cycle 14, and none of them completed embryogenesis (Supplementary Fig. 4d)[24,27]. This observation goes in line with previous reports showing a complete arrest of embryogenesis at the ZGA stage in CBP perturbed embryos across species, strengthening its conserved role in regulating early embryogenesis[24,31,89–91].

To investigate whether any maternal effects upon CBP or E(z) depletion could cause defects in embryogenesis before ZGA, we performed bulk RNA-seq in embryos before cycle 9 and at cycle 14 (Supplementary Fig. 4e). Notably, Principal component analysis (PCA) showed mild transcriptomic differences between wild-type and maternally depleted embryos before cycle 9, suggesting that the germline depletion of these factors does not drastically impair the maternal RNA load. In contrast, a marked transcriptomic difference was observed at cycle 14, confirming that the depletion of E(z) or CBP has a direct effect on the ZGA process.

Moreover, in agreement with the immunofluorescence, chromatin from E(z)-KD and CBP-KD cycle 14 embryos showed reduced levels of the corresponding mark, as assessed by bulk CUT&Tag (Supplementary Fig. 4f–h).

## E(z) and CBP play distinct roles in regulating cell identities in early embryos

To characterize the roles of CBP and E(z) at single-nucleus resolution, we performed 10× Multiome across two biological replicates in precisely hand-selected E(z)-KD and CBP-KD embryos at ZGA (Fig. 3a). After quality filtering and data integration of wild-type and KD datasets, we retrieved a total of 22,358 nuclei. UMAP dimensional reduction

based on the RNA embedding highlighted transcriptomic differences between WT and mutants (Fig. 3b, left panel, and Supplementary Data 4). By using the previously mentioned marker genes for annotation (as shown in Supplementary Fig. 2b and Supplementary Data 2), we could successfully identify the main germ layers in E(z)-KD embryos but not in the CBP-KD counterparts (Fig. 3b, right panel and Supplementary Data 5). By using either scATAC-seq or combined scATAC-seq + scRNA-seq data, we confirmed that chromatin accessibility also contributes to the differences between wild-type and KD conditions as well as germ layers, as highlighted by very well-defined clusters in both UMAP embeddings (Supplementary Fig. 5a–c).

Quantification of the relative wild-type and mutant cell fractions per germ layer revealed a significant loss of cell identity in embryos depleted of either chromatin factor. E(z) mutants exhibited a redistribution of nuclei across annotated germ layers (Fig. 3c and Supplementary Data 5), with a gain of yolk and undifferentiated cells when compared to wild-type embryos. Strikingly, embryos depleted of the co-activator CBP presented a more dramatic loss of cellular identity, with most cells classified as yolk and two undifferentiated clusters that did not show enrichment for germ layer marker gene signatures (Supplementary Fig. 5d). This prompted us to investigate whether CBP depleted nuclei might exhibit an increase tendency towards pluripotency by employing an entropy score analysis tool[92]. Briefly, using a protein-protein interaction (PPI) network, the number of active biological processes per genotype is measured. Since a committed or differentiated cell preferentially activates specific pathways, it would manifest a lower entropy rate. On the contrary, pluripotent cells would show a state of uncertainty or promiscuous signaling and, thus, high entropy levels. Our analysis revealed a higher entropy score in the CBP-KD condition, likely due to the undifferentiated clusters, underscoring the essential function of CBP in promoting pluripotency exit during embryogenesis (Supplementary Fig. 5e).

Taken together, our data suggest the fundamental and distinct roles of E(z) and CBP in establishing and maintaining cellular identity during ZGA.

## H3K27me3 protects germ layer-specific genes from aberrant CBP-mediated activation

To investigate in detail how cellular identities change upon loss of each chromatin factor, we examined the single-cell changes in gene expression and chromatin accessibility in E(z)-KD and CBP-KD embryos compared to wild-type condition.

E(z) depleted embryos displayed an overall misexpression of marker genes across different germ layers, which was accompanied by a widespread increase in chromatin accessibility at both promoters and enhancers regions (Fig. 3d, Supplementary Fig. 5f and Supplementary Data 6). In particular, known markers of the dorsal ectoderm

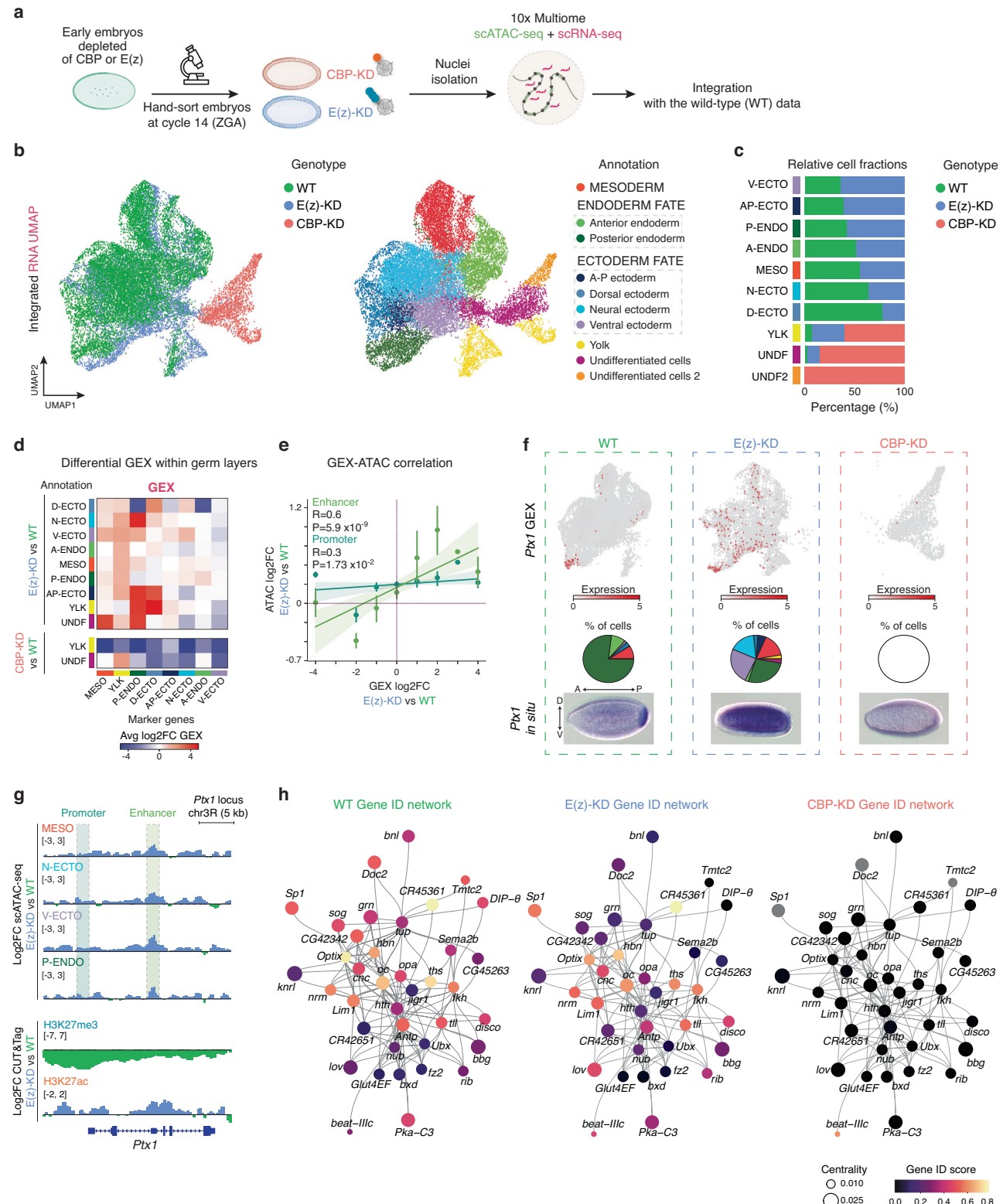

**a** Early embryos depleted of CBP or E(z) → Hand-sort embryos at cycle 14 (ZGA) → CBP-KD / E(z)-KD → Nuclei isolation → 10x Multiome scATAC-seq + scRNA-seq → Integration with the wild-type (WT) data

**b** Integrated RNA UMAP

Genotype: WT, E(z)-KD, CBP-KD

Annotation: MESODERM; ENDODERM FATE (Anterior endoderm, Posterior endoderm); ECTODERM FATE (A-P ectoderm, Dorsal ectoderm, Neural ectoderm, Ventral ectoderm); Yolk; Undifferentiated cells; Undifferentiated cells 2

**c** Relative cell fractions

**d** Differential GEX within germ layers

**e** GEX-ATAC correlation. Enhancer R=0.6, P=5.9 ×10⁻⁹; Promoter R=0.3, P=1.73 ×10⁻²

**f** WT / E(z)-KD / CBP-KD. Ptx1 GEX; Ptx1 in situ

**g** Ptx1 locus chr3R (5 kb). Log2FC scATAC-seq E(z)-KD vs WT; Log2FC CUT&Tag E(z)-KD vs WT

**h** WT Gene ID network; E(z)-KD Gene ID network; CBP-KD Gene ID network

and posterior endoderm became upregulated and more accessible in multiple other embryonic tissues. Interestingly, changes in gene expression upon E(z)-KD correlated more strongly with differential opening of enhancers rather than promoters, supporting the role of E(z) in preserving proper germ layer transcription by tuning chromatin accessibility (Fig. 3e). Notably, E(z) depletion caused aberrant deposition of H3K27ac (Supplementary Fig. 5g), in line with previously

published data across different species, thus highlighting its conserved role[27,55,93,94].

Conversely, CBP appears to be essential for ZGA, with its absence leading to a complete cessation of transcription (Fig. 3d, Supplementary Fig. 3f and Supplementary Data 6). Interestingly, transcriptional shutdown was not followed by a reduction in chromatin accessibility in neither promoters nor enhancers upon CBP depletion (Supplementary

**Fig. 3 | Loss of cell fate occurs upon chromatin factors depletion.**
**a** Experimental design. Early embryos depleted of CBP or E(z) are aged and hand-selected at ZGA based on their morphology. Subsequently, nuclei isolation is performed and 10× Multiome (scATAC-seq + scRNA-seq) is performed in two biological replicates. Adapted from an image created in BioRender. Cardamone, F. (2025) https://BioRender.com/a06c492. **b** UMAP embedding of integrated scRNA-seq of wild-type, E(z)-KD and CBP-KD data. Left, cluster identities were assigned based on the respective genotype. Right, cluster identities were assigned by expression of marker genes. **c** Bar plot showing the relative abundance per annotated germ layer. To calculate the relative abundance, the total number of cells in each genotype was considered. UNDF, undifferentiated cells. UNDF-2, undifferentiated cells 2 (CBP-KD specific). **d** Average log2 fold change of marker gene expression upon E(z)-KD or CBP-KD. The heat map shows ectopic expression of marker genes upon loss of E(z) or complete shut-down of transcription upon CBP-KD. **e** Correlation plot between differential GEX and accessibility of promoters

(dark green) or enhancers (light green) of marker genes upon E(z)-KD in each single cell. Spearman correlation coefficient (R) and *P*-value (P) are shown. Data are presented as mean values +/- SD. **f** Top, Gene expression (GEX) of *Ptx1* (posterior endoderm marker gene) in wild-type, E(z)-KD and CBP-KD. Pie charts represent fraction of cells expressing the gene within each germ layer. Bottom, Representative in situ hybridization image from three biological replicates of *Ptx1* RNA in wild-type, E(z)-KD and CBP-KD. A anterior, P posterior, D dorsal, V ventral. **g** Top, Genome browser snapshot of *Ptx1* locus. Differential accessibility between E(z)-KD and wild-type cells in different germ layers. Promoter (P) is highlighted with a blue dashed box while enhancer (E) is highlighted with a green dashed box. Bottom, Differential CUT&Tag signal between E(z)-KD and wild-type for H3K27me3 and H3K27ac. **h** Gene identity network of wild-type, E(z) and CBP depleted embryos. Gene ID score refers to the ratio of expressing cells which are annotated as the expected germ layer, with respect to the total number of cells.

Fig. 3f and Supplementary Data 6), as also described in other cellular contexts or model organisms[95–97].

To exemplify the consequences of E(z)-KD and CPB-KD more deeply, we focused on two specific genes: *Ptx1*, a paired-like homeobox transcription factor[98], and *Doc2*, a tissue-specific T-box transcription factor[99], which are markers of posterior endoderm and dorsal ectoderm, respectively. Both marker genes were ectopically expressed in most germ layers in E(z)-KD embryos, whereas their expression was absent in CBP-KD embryos, as also confirmed by in situ hybridization (Fig. 3f and Supplementary Fig. 6a). Furthermore, *cis*-regulatory elements of both *Ptx1* and *Doc2*[36] were aberrantly accessible in multiple different cell types upon depletion of E(z) (Fig. 3g and Supplementary Fig. 6b–d). Moreover, we observed increased H3K27ac enrichment at both loci upon H3K27me3 depletion induced by E(z)-KD, which was associated with a mild elevation of CBP binding and increased transcript levels, in contrast to CBP-KD embryos (Supplementary Fig. 6e, f).

Based on these results, we interrogated how our previously inferred wild-type GRN of the ZGA embryo would be affected upon perturbing the chromatin landscape by E(z) or CBP depletion (Fig. 2j). Interestingly, E(z) depletion caused a loss of cell identity also in key TFs involved in germ layer specification, which become ectopically expressed across cell types (Fig. 3d–h). Notably, while some TFs remained expressed in the expected germ layer, such as *Lim1* or *oc*, others, like *opa*, became broadly expressed across multiple cell types. This aberrant expression of key TFs may disrupt the underlying regulatory networks specific to each germ layer. In contrast, CBP depletion resulted in the complete loss of TF identity, leading to a failure in germ layer specification (Fig. 3c and Supplementary Fig. 5d).

The findings collectively suggest that E(z)-dependent H3K27me3 determines the identity of different germ layers by selectively modulating the aberrant activation of enhancers of essential marker genes by CBP and H3K27ac. Conversely, CBP functions as a fundamental transcriptional activator of the embryonic genome, as its absence results in a complete cessation of transcription without affecting chromatin accessibility.

## Canonical enhancer activity patterns are lost upon chromatin factors depletion

Our findings suggest that the chromatin environment surrounding enhancer regions must be finely tuned to achieve the correct gene activation/repression in a cell type-specific fashion. To validate this result, we generated transgenic fly lines to test the in vivo expression pattern of enhancers that were identified both in our scATAC-seq peaks and in a curated database (Supplementary Fig. 6c–f)[80]. Specifically, we examined two previously characterized regulatory regions for *Ptx1* and two for *Doc2*[36], in control embryos as well as embryos depleted of E(z) or CBP during ZGA (see Methods and Supplementary Fig. 7a).

We analyzed the reporter expression patterns by in situ hybridization using a probe against LacZ. As expected, control embryos displayed canonical expression patterns of the reporter gene[36]. Interestingly, E(z) depleted embryos exhibited a spreading of the reporter expression areas compared to control, indicating aberrant enhancer activity (Supplementary Fig. 7b). This observation supports our hypothesis that H3K27me3 restricts enhancer activity to the appropriate cell types. For example, an ectopic reporter expression for *Doc2* enhancers was observed in the anterior-posterior axis and ventral ectoderm of the embryo, consistent with our multiomic results (Fig. 3f, g, Supplementary Fig. 5f and Supplementary Fig. 6a,f). However, the reporter lines did not fully recapitulate the severe phenotypes observed in the in situ hybridization experiments for the two endogenous genes upon E(z) depletion (Fig. 3f and Supplementary Fig. 6a). This suggests that the loss of H3K27me3 may result in the combinatorial activation of multiple enhancers, rather than the misregulation of a single regulatory region, driving ectopic gene expression. In contrast, CBP-depleted embryos displayed a complete loss of the expression pattern, mimicking the results obtained for the gene expression of *Ptx1* and *Doc2*, confirming the role of CBP as an activator of developmental enhancers (Fig. 3f, Supplementary Fig. 6a and Supplementary Fig. 7b).

Together, these in vivo reporter assays build on our previous findings and suggest that strict chromatin regulation is essential during early embryogenesis to ensure the proper activation or repression of developmental enhancers at key regulatory genes during ZGA.

## Changes in the chromatin landscape affect the developmental dynamics of embryogenesis

To assess how perturbations in the epigenetic landscape, resulting from CBP or E(z) depletion, influence the timing of embryogenesis, we leveraged a previously published single-cell epigenomic and transcriptomic *continuum* of *Drosophila* development[56]. We selected from this reference six time points to capture embryonic stages from before ZGA to gastrulation, with 0–2 and 1–3 h classified as early stages, 2–4 and 3–7 h as middle stages, and 4–6 and 8–10 h as late stages (Fig. 4a). After projecting our wild-type and mutant single-cell multiome data into this reference (Fig. 4b), we analyzed the distribution of each genotype across embryogenesis.

As expected, wild-type nuclei predominantly aligned with the middle clusters for both gene expression and chromatin accessibility, especially enriched in the 3 to 7-h group (Fig. 4c, d and Supplementary Fig. 7c). E(z)-KD showed a marginal shift towards the early cluster both in transcriptomic and chromatin accessibility dimensions. On the other hand, CBP depleted embryos showed the most pronounced deviation, with all cells enriching in the early stage cluster based on the transcriptome, indicating a dramatic transcriptional delay upon CBP depletion. In line with this, cell fate specification of the different germ layers is fully achieved in the mid time point, corresponding to the ZGA

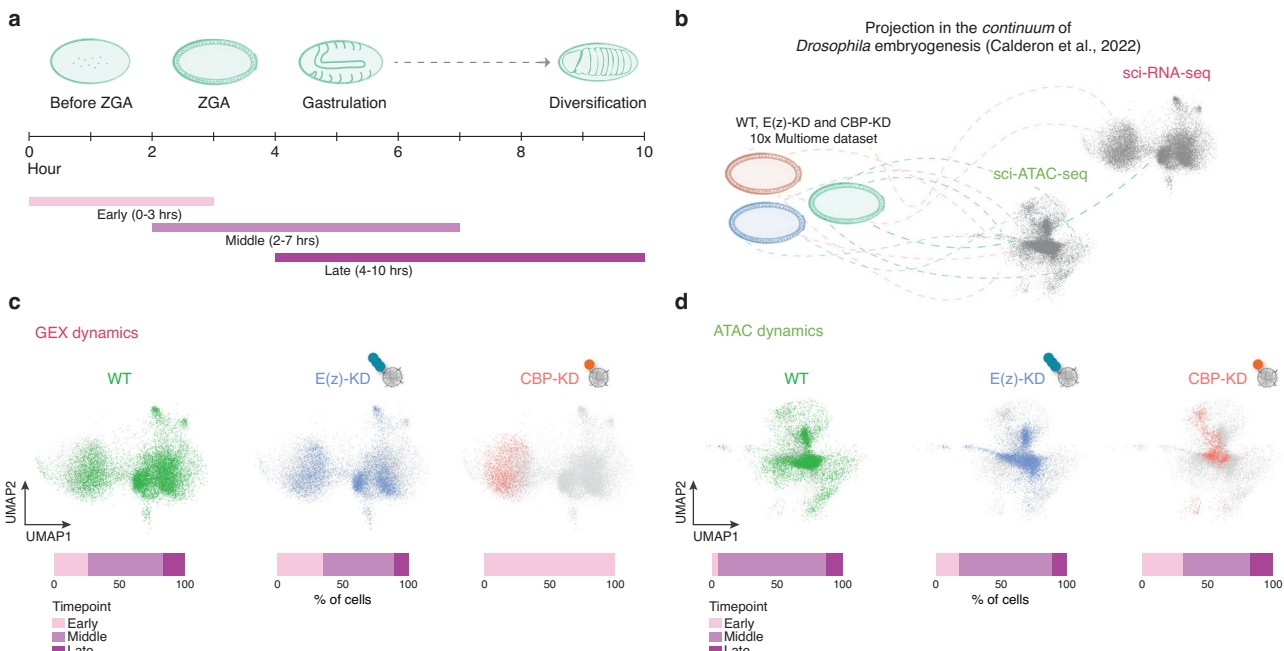

**Fig. 4 | Transcriptomic and epigenomic temporal dynamics upon chromatin factors depletion reveal transcription-independent chromatin progression.** **a** Schematic of early *Drosophila* development, from fertilization to diversification events. Early, middle and late time points are highlighted in pink, light purple and dark purple respectively. **b** Analysis design. 10× Multiome data from wild-type, E(z)-KD and CBP-KD nuclei at ZGA were projected in the sci-RNA-seq or sci-ATAC-seq *continuum* of *Drosophila* embryogenesis[56] to assess the developmental progression of each projected genotype for both transcription and chromatin accessibility. **c** UMAP embedding of the scRNA-seq (GEX) data colored by genotype with respective bar plot representing the fraction of nuclei enriched in each developmental time point. The projection shows the developmental delay caused by the absence of CBP. **d** Same as (**c**) but for scATAC-seq peaks (ATAC). The projection highlights the dispensable function of CBP in establishing global accessibility.

stage, while early clusters were predominantly enriched by undifferentiated and yolk cells that were mostly present in CBP-KD (Supplementary Fig. 7e).

Interestingly, chromatin accessibility was largely maintained in CBP-KD embryos, suggesting that despite the absence of CBP, H3K27ac, and transcription, chromatin accessibility progressed in a manner comparable to that in control nuclei (Fig. 4c, d and Supplementary Fig. 7c). This suggests that regulatory mechanisms governing chromatin accessibility in these early embryos may differ from those found in other organisms or in more differentiated somatic cells, where CBP depletion typically causes both transcriptional shutdown and reduced chromatin accessibility[50,100]. Consistent with this, accessibility at promoters and enhancers remained largely unchanged, indicating that CBP primarily regulates their activation rather than their accessibility, as further supported by our in vivo enhancer reporter assay (Supplementary Fig. 5f, Supplementary Fig. 6c–f and Supplementary Fig. 7b).

To validate our data with an orthogonal approach, we computationally conducted a pseudo-time analysis using our integrated single-cell ZGA dataset (Supplementary Fig. 7f, g). This analysis confirmed a transcriptomic delay in CBP-depleted embryos, whereas E(z)- depleted nuclei progressed further in embryogenesis similarly to wild-type. According to pseudotime ordering, the first germ layers to be defined were the posterior endoderm, ventral ectoderm, and anterior endoderm, followed by the dorsal to neural ectoderm specification, with mesoderm nuclei lastly acquiring their identity (Supplementary Fig. 7h). Notably, these results are in line with our projection in the 1–3 h window (before ZGA) of the real-time analysis, where only anterior-posterior endoderm and ventral ectoderm cells could be detected (Supplementary Fig. 7e, h). Interestingly, depletion of E(z) and H3K27me3 resulted in a mild transcriptional delay, leading to alterations in germ layer proportions. This aligns with our hypothesized specification timing, as germ layers that are formed earlier (such

as posterior endoderm, undifferentiated, and yolk cells) expanded compared to wild-type, while later forming tissues, including the neural and dorsal ectoderm, were partially reduced (Fig. 3c and Supplementary Fig. 7d).

In summary, our temporal analysis underscores the role of chromatin factors, particularly CBP, in regulating both cell fate commitment and the precise timing of cell fate specification during early embryogenesis.

## CBP stabilizes pioneer factor binding and initiates zygotic genome activation

Next, we aimed to address whether the severe loss of cell fate upon CBP depletion could be attributed to the deregulation of pioneer factors by using CUT&Tag (Supplementary Fig. 8a).

To do so, we used our previously characterized list of actively transcribed genes at cycle 14 identified by GRO-seq, annotated by their dependency on the pioneer factor Zelda or the histone variant His2Av, hereafter referred to as H2A.Z[23,101]. Interestingly, Zelda-dependent genes showed high levels of H3K27ac before cycle 9 and ZGA, whereas H2A.Z-negative and -positive genes gained H3K27ac around cycle 10–12 and concomitantly with the minor wave of ZGA, at both promoters and enhancers (Fig. 5a and Supplementary Fig. 8b, c). The three gene categories exhibited high levels of Zelda, GAGA factor (GAF), and CBP binding during cycle 14 under physiological conditions. Interestingly, in CBP-KD embryos, we observed an overall reduction of Zelda and GAF binding at both promoters and enhancers, suggesting that CBP might be required to stabilize pioneer factors binding on chromatin (Fig. 5a and Supplementary Fig. 8b–d). Moreover, qPCR and bulk RNA-seq experiments suggest that CBP depletion does not directly affect the RNA levels of Zelda and GAF (Supplementary Fig. 8e). These findings are consistent with our previous report, which showed no significant changes in the protein levels of these two pioneer factors upon CBP depletion[24].

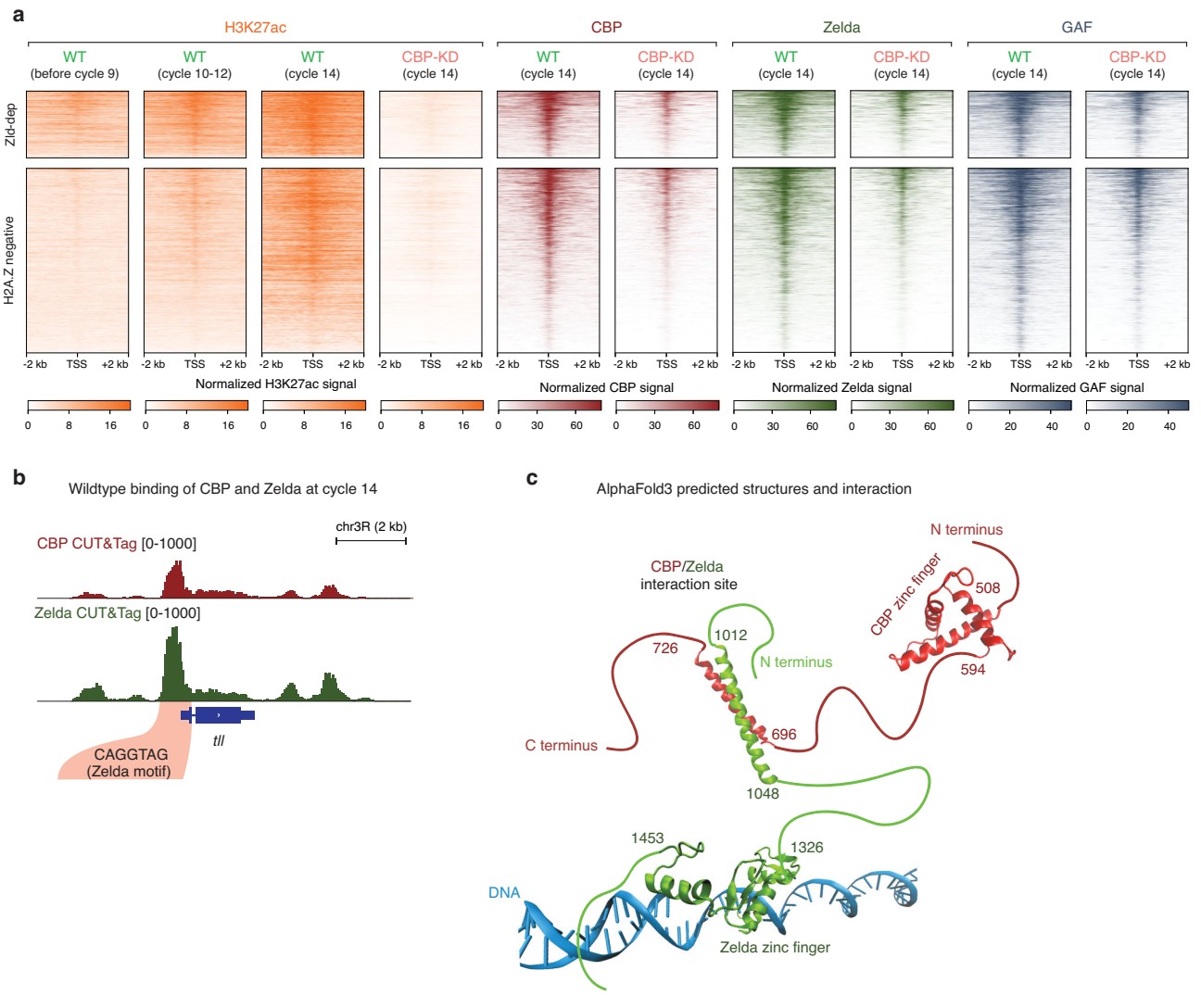

**Fig. 5 | CBP stabilizes pioneer factors binding on chromatin through a possible coiled coil interaction. a** Heat map of active ZGA genes identified by GRO-seq, clustered by their dependency on the pioneer factor Zelda (Zld-dep) or neither regulated by Zelda or H2A.Z (H2A.Z negative)[23] Signal is centred around +/− 2 kb from TSS. Sorting is based on H3K27ac levels before cycle 9 and shows CUT&Tag signal distribution of this mark at three developmental time points and signal distribution of CBP, Zelda and GAF at ZGA in both wild-type and CBP-KD at cycle 14. **b** CUT&Tag signal of CBP and Zelda at *tll* locus, where both factors are bound during cycle 14 in wild-type embryos. The highlighted region indicates the localization of the Zelda motif used to predict the possible structure of binding between Zelda and CBP. **c** Alphafold3[102] predicted structure and interaction between CBP and Zelda. Secondary structures are predicted by Alphafold3 and relaxed by molecular dynamics simulations of the DBD or coiled coil regions of CBP and Zelda. The amino acid ID is indicated from N terminus to C terminus. The red and green lines sketch the protein structure of CBP and Zelda, respectively. DNA of *tll* locus where Zelda motif is present is coloured in blue.

In addition, expression of Zelda-dependent and H2A.Z-negative genes was reduced in CBP depleted embryos compared to wild-type or E(z)-KD, whereas H2A.Z-positive genes were unaffected since they are involved in regulating housekeeping transcriptional programs by the histone chaperon Domino and the histone variant H2A.Z (Supplementary Fig. 9a)[23]. Gene ontology analysis of the affected Zelda-dependent and H2A.Z-negative genes revealed enrichment in terms associated with cell fate commitment (Supplementary Fig. 9b).

To explore the potential mechanism by which CBP might stabilize Zelda, we used AlphaFold3[102] to model the structures and possible interactions between these proteins. For this analysis, we selected a locus containing the Zelda DNA binding motif that showed co-occupancy of CBP and Zelda in CUT&Tag experiments from wild-type cycle 14 embryos (Fig. 5c). Structurally, both Zelda and CBP were predicted to contain many intrinsically disordered regions (IDRs)

(Supplementary Fig. 9c). Within these regions, the amino acid residues 696–726 of CBP and 1012–1048 of Zelda were predicted to have a high propensity for forming coiled coil motifs, which are often associated with protein-protein interactions. Alpfafold3 modeling suggested a possible interaction between these coiled coil regions, yealding a predicted dimeric complex with a calculated binding free energy of −6.08 kcal/mol. Based on this prediction, this coiled coil interaction could theoretically tether the zinc-finger of Zelda (residues 1326 to 1435) and of CBP (residues 508–594), constraining a spatial arrangement to a maximum distance of 144 nm. This hypothetical positioning might facilitate Zelda's zinc finger binding to its DNA binding motif at the *tll* promoter region (Fig. 5c).

However, it is important to note that these AlphaFold3 predictions are purely computational and remain non-validated experimentally. We cannot rule out the possibility that CBP interacts with Zelda

indirectly through an intermediate protein. Additional experimental studies will be required to confirm or refine these hypotheses and to uncover the precise mechanism by which CBP might influence Zelda function.

## CBP regulates RNA Polymerase II dynamics during ZGA

Our results suggest that CBP depletion does not lead to significant chromatin compaction despite the dramatic shutdown of transcription and destabilization of pioneer factor binding (Fig. 5a, Supplementary Fig. 5d–f, Supplementary Fig. 6a–f, Fig. 4c, d and Supplementary Fig. 7b–f). Based on this, we hypothesized that, in addition to the residual pioneer factor binding still associated with chromatin (Fig. 5a and Supplementary Fig. 8b, c), RNA polymerase II (RNAPII) may also remain bound to chromatin upon CBP loss, which could help prevent chromatin compaction during ZGA[50,83,103,104].

To investigate CBP's role in transcriptional regulation, we performed CUT&Tag on carefully hand-selected cycle 14 (ZGA) wild-type and CBP-KD embryos, targeting RNAPII phosphorylations at serine 5 (RNAPII-S5P) and serine 2 (RNAPII-S2P), which correspond to the initiation and elongation forms of RNAPII, respectively (Supplementary Fig. 10a). Focusing on nascent GRO-seq genes at ZGA, categorized by Zelda dependency[23,101], we observed reduced RNAPII-S5P levels, particularly at the promoters of Zelda-dependent genes in CBP-depleted embryos (Fig. 6a). This finding suggests that Zelda destabilization might contribute to impaired RNAPII recruitment at these regions.

In contrast, RNAPII-S2P levels were globally reduced across gene bodies of both Zelda-dependent and independent gene subsets (Fig. 6a). As a result, the pausing index increased upon CBP-KD, correlating with the decreased binding of the bromodomain protein BRD4, which is an important regulator of the pause-release process[105–107] (Fig. 6b and Supplementary Fig. 10b–d). Moreover, non transcribed genes at ZGA (GRO-seq negative) displayed no RNAPII-S5P or RNAPII-S2P signals (Fig. 6a). These results highlight the essential role of CBP in regulating transcriptional elongation (Fig. 6a), consistent with previous reports in S2 cells[103].

Interestingly, we observed an accumulation of paused RNAPII-S5P at enhancers of germ layer specific genes. This suggests that paused RNAPII may play a role in maintaining chromatin accessibility at both enhancers and promoters, alongside the partially retained binding of pioneer factors and BRD4, even in the absence of CBP and active gene expression (Fig. 6c and Supplementary Fig. 6e).

In conclusion, our data strongly suggest that transcriptomic dynamics and chromatin accessibility are perturbed upon E(z) and CBP depletion, resulting in post-ZGA lethality. E(z)-mediated H3K27me3 safeguards cell type-specific transcription by preventing ectopic H3K27ac deposition and aberrant chromatin accessibility. Conversely, CBP plays a crucial role in determining cell fate by stabilizing pioneer factors binding on chromatin and promoting transcriptional elongation, which ensures the proper activation of germ layer-specific genes (Fig. 7).

## Discussion

Addressing the epigenetic contribution of chromatin architecture to the *cis*-regulatory code presents a significant challenge, especially when related to cell fate specification in vivo[108]. In our study, we explored the transcriptomic-epigenomic interplay that orchestrates the molecular dynamics of early *Drosophila* development before and at ZGA with single-cell resolution. We discovered two complementary mechanisms that govern cell fate specification: (i) E(z)-dependent H3K27me3 prevents ectopic gene expression by modulating enhancer activation of germ layer-specific genes, and (ii) CBP regulates transcriptional activation of key developmental genes at ZGA by stabilizing pioneer factors binding and regulating transcriptional elongation.

We found that histone modifications on chromatin precede the active transcription of crucial genes involved in cell fate determination. This indicates a major chromatin remodeling at the very start of embryogenesis, before ZGA, highlighting the pivotal role of chromatin in regulating embryonic tissue specification. The dominant role of H3K27me3 is consistent with evidence that this modification is inherited from the maternal germline in both *Drosophila* and mammals[27,30], thus preceding the de novo establishment of H3K27ac.

We provide a comprehensive in vivo scRNA-seq and scATAC-seq Multiome of the ZGA *Drosophila* embryo, simultaneously profiling epigenomic and transcriptomic information from the same nuclei, which is fully explorable at https://iovinolab.shinyapps.io/scmultiomeZGA/. We uncover that the precursors of various germ layers are already delineated at ZGA at both transcriptomic and epigenomic levels, preceding any morphogenetic changes. These results indicate that the foundational steps of cell fate determination are set in motion before gastrulation and highlight the major role played by epigenetic mechanisms, already acting at this early stage and being required to establish and preserve cellular identities. Notably, we show that accessibility at enhancer regions serves as a more reliable indicator of cell-specific transcription than accessibility at promoters.

Our results underscore the crucial role of chromatin in regulating germ layer-specific enhancers and orchestrating early developmental processes. Maternal depletion of E(z) or CBP, along with their respective modifications, led to a loss of cellular identity, each with distinct outcomes as revealed by single-cell multiomic analysis. Specifically, the depletion of E(z) and H3K27me3 resulted in increased chromatin accessibility at key *cis*-regulatory elements. This aberrant accessibility triggered spurious activation of enhancers by CBP and other tissue-specific TFs, ultimately leading to ectopic expression of marker genes across different germ layers. Conversely, the absence of CBP led to transcriptional arrest and reduced chromatin occupancy of Zelda and GAF, resulting in loss of zygotic transcription and cellular commitment. By using AlphaFold predictions, we identified a potential interaction between CBP and Zelda via coiled coil regions, consistent with the evidence of their colocalization in discrete clusters immediately preceding transcription initiation[109]. However, this interaction remains speculative, and additional experiments are required to validate it. Notably, CBP intrinsically disordered regions (IDRs) have been reported to mediate the stabilization of various transcription factors, supporting its proposed role as a scaffold during active transcription[110–115]. Future studies will be necessary to further elucidate CBP's role in pioneer factors stabilization and whether other structured domains of CBP could play a role in such process.

Interestingly, although CBP depletion arrested cells in an undifferentiated state with a transcriptional block at the pre-ZGA stage, chromatin accessibility continued to mature in CBP-KD nuclei. Our analysis revealed that the loss of CBP led to a global increase in the paused RNAPII state, consistent with previous in vitro observations[50,103]. This pausing likely contributed to maintaining chromatin accessibility at both enhancers and promoters of key developmental genes, supported by the partial retention of pioneer factors, even in the absence of active transcription.

Taken together, we speculate that upon depletion of CBP, chromatin accessibility is affected during the first phase of a two-step process[116]. Initially, enhancers become accessible due to the binding of transcription factors. These enhancers remain in a waiting state until the appropriate combination of co-factors (such as CBP) or additional transcription factors are bound and stabilized by CBP to subsequently activate the zygotic genome[96,104,115]. This highlights the importance of both pioneer factors and chromatin activators in initiating cell type-specific transcriptomic networks in the early embryo.

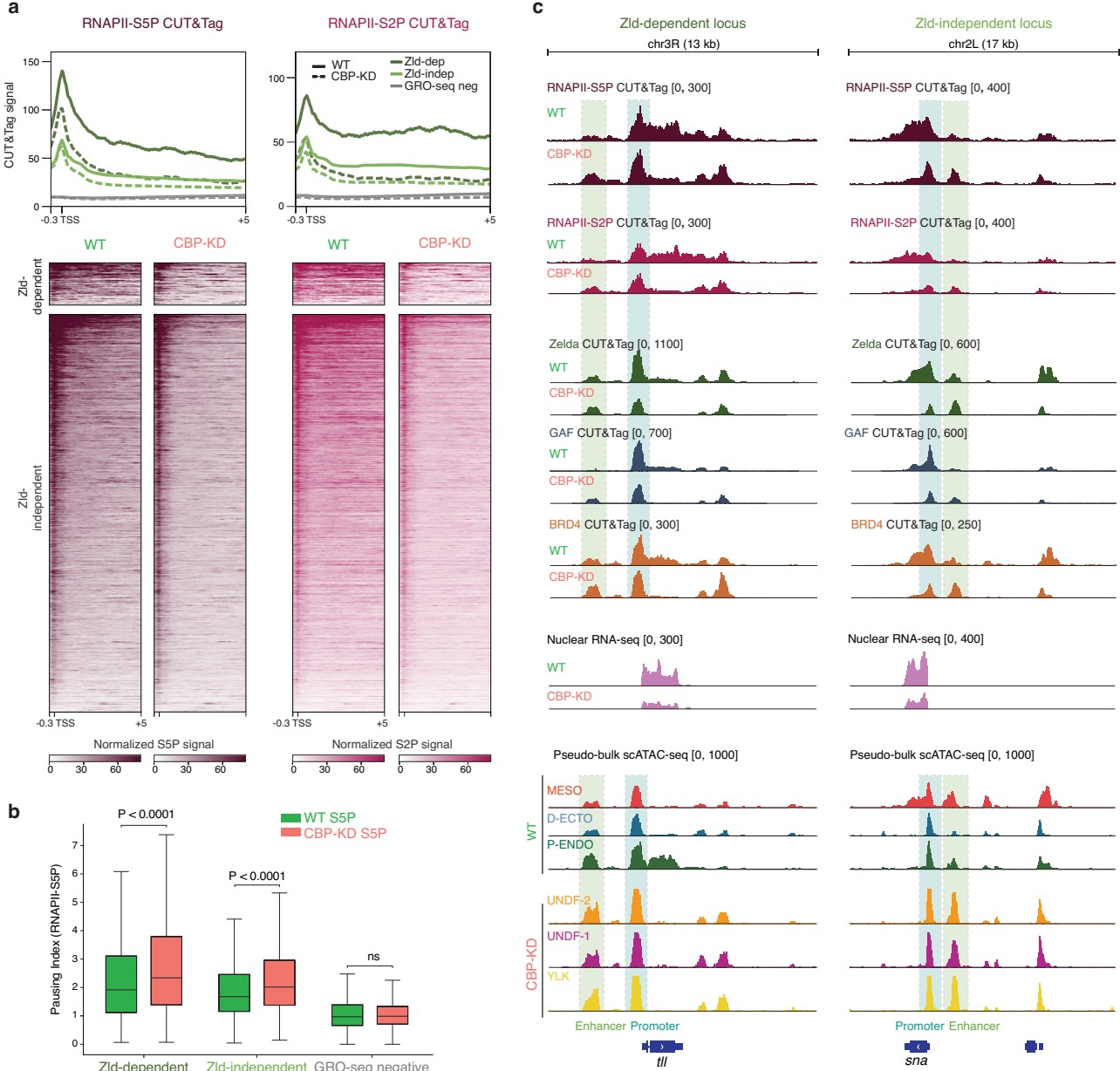

**Fig. 6 | CBP depletion impacts RNA Polymerase II dynamics by increasing its pausing state at zygotically transcribed genes. a** Profile plot and heat map of active ZGA genes identified by GRO-seq[23], classified as Zelda-dependent (Zld-dep), Zelda-independent (Zld-independent) according to[101], or non-transcribed genes (GRO-seq negative). Signal is centred around −300 bp/+5 kb from TSS. Sorting is based on GRO-seq levels and shows CUT&Tag normalized signal distribution of RNAPII-S5P and -S2P, respectively initiated and elongated isoforms of RNA Polymerase II, in both wild-type and CBP-KD at cycle 14. **b** Pausing index of RNAPII-S5P at Zelda-dependent, Zelda-independent and GRO-seq negative genes in both wild-

type and CBP-KD condition. Boxes center refers to mean, lower and upper quartiles (Q1 and Q3, respectively). Whiskers, 1.5 × IQR below Q1 and above Q3. Two-sided Mann−Whitney $U$ test. Zelda-dependent genes, $P = 6.010 \times 10^{-08}$; Zelda-independent genes $P = 3.141 \times 10^{-35}$. $n = 2$ biological replicates per condition. **c** Genome browser snapshot of Zelda-dependent (*tll*) or Zelda-independent (*sna*) loci. CUT&Tag, nuclear RNA-seq and aggregated scATAC-seq for germ layers in wild-type (WT) and CBP-KD data are displayed. Promoter region is highlighted by a blue box, enhancer region is highlighted by a green box.

In conclusion, our study highlights the essential role of chromatin modifiers in regulating early developmental processes in *Drosophila*, setting the stage for successful gastrulation. Our findings demonstrate that these modifiers coordinate the repression or activation of cell type-specific enhancers, ensuring the precise timing of transcriptional and epigenetic dynamics in the zygote. We propose a potential crosstalk between E(z) and CBP, where maternally inherited H3K27me3 may be eroded through the combined action of chromatin remodelers and pioneer transcription factors alongside the recruitment of CBP to promote active transcription. The H3K27-specific

demethylase *Utx*, which has been shown to interact with the chromatin-remodeler Brahma and CBP itself[117], could contribute to this process. Through such mechanisms, different cell types derived from the same progenitor could modulate the activation and repression of specific transcriptional programs, depending on whether the same regulatory element is in a repressive or active state.

## Methods

Further information and requests for materials, resources and reagents should be addressed to Yinxiu Zhan or Nicola Iovino.

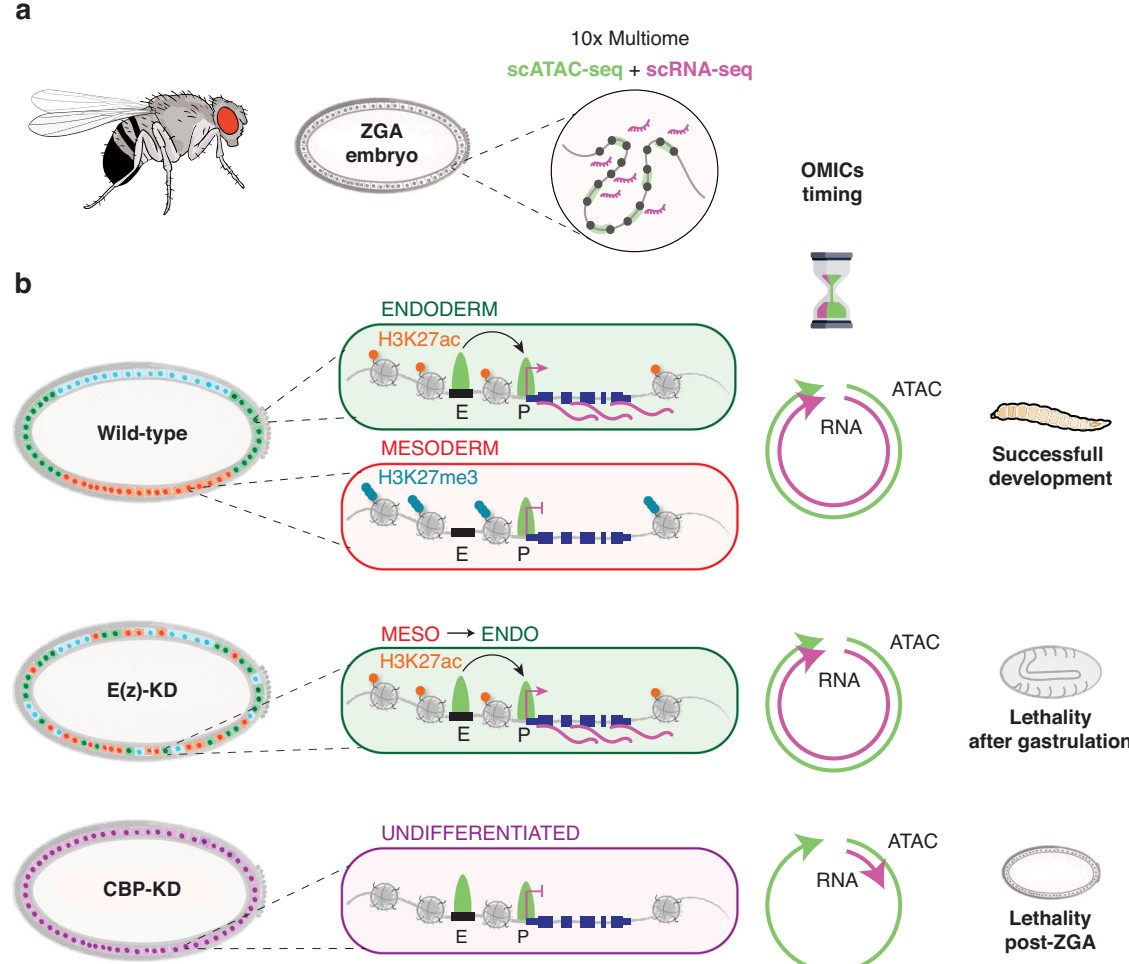

**Fig. 7 | Multiomic dynamics of germ layer specification during ZGA in *Drosophila* embryos. a** Schematic of the 10× Multiome (scRNA-seq + scATAC-seq) profiling of the *Drosophila* ZGA embryo. **b** Simultaneous dissection of the chromatin landscape and transcriptomic state from each single nucleus provide a comprehensive overview of the dynamics shaping every cell of the embryo. Differential enrichment of H3K27me3 and H3K27ac at the same loci can shape different transcriptomic outputs by coordinating enhancer activation or repression, providing the blueprint for gastrulation and progression in development. Chromatin factor depletion affects enhancer regulation in two ways: depletion of maternal E(z) and H3K27me3 activate aberrant enhancers across cell-types, causing ectopic gene expression. Instead, CBP loss causes a massive block of differentiation given by the halted transcription at the pre-ZGA stage, while chromatin accessibility can still progress. In both scenarios, embryogenesis cannot be completed. E, Enhancer. P, Promoter. Adapted from an image created in BioRender. Cardamone, F. (2025) https://BioRender.com/s77z537.

## Fly stocks

The fly lines employed in this manuscript were raised on corn flour molasses medium (consisting of 12 g agar, 18 g dry yeast, 10 g soya powder, 22 g molasses, 80 g malt extract, 80 g corn flour, 6.25 ml propionic acid, and 2.4 g nipagin per litre of water) at a temperature of 25 °C. TRiP line control (BDSC, #36303), female germline-specific Gal4-driver P{matα4-GAL-VP16}V37 (BDSC, #7063), Nejire shRNA#1 (CBP homemade line)[24], E(z) shRNA P{TRiP.HMS00066}attP2 (BDSC, #33659). A complete list of the fly stocks used in this study is available in Supplementary Data 7.

## Induction of RNAi in the female germline and embryonic phenotypical characterization

To generate maternal knockdown (KD) via RNA interference (RNAi), we adopted the Gal4-UAS system. Briefly, RNAi was induced by mating females containing a transgene that encodes a short hairpin RNA (shRNA) with males carrying the Gal4-driver at 25 °C. See Supplementary Data 7 for detailed information on the fly lines.

The F1 generation resulting from these crosses exhibits KD for a specific target in the female germline and subsequent egg laying by these females (F2 generation) produces offspring lacking the particular transcript. This strategy ensures that late-stage ovaries are free of the target mRNA and protein without disrupting meiosis or fertilization, facilitating the examination of early embryos lacking both mRNA and the respective product. To determine the hatching rate of control and RNAi mediated KD F2 embryos, flies were let laying for 0–1 h on apple juice agar plates and 100 embryos were randomly selected and aligned in group of 10 in a new plate. Embryos were classified as "cellularized" upon reaching stage 5 of embryogenesis (nuclear cycle 14), characterized by a distinct thick rim at the periphery of the embryonic syncytium and complete invagination of cellular membranes. The Cellularization Rate was determined by the proportion of cellularized embryos among the total aligned embryos. Subsequently, embryos reaching the L1 larval stage after 25 h from alignment were categorized as "hatched" and Hatching Rate was scored. Both Cellularization and Hatching rate were calculated by averaging from a minimum of three biological replicates. A second count was performed after 48 h to ensure that hatching was not occurring later in time.

## Embryo immunofluorescence

Control and KD embryos were collected separately for 0–30 min, 0–120 min and 0–240 min to cover approximatively all the early stages of embryogenesis. Successively, embryos were dechorionated in 50% bleach for 2 min, fixed with ice-cold methanol:heptane (1:1). After, embryos were vigorously shacked with the help of a vortex to facilitate devitellinization process. The sedimented devitellinized embryos were collected in a fresh 1.5 ml Eppendorf tube and underwent three subsequent washes with ice-cold methanol. Devitellinized and fixed embryos underwent rehydration and permeabilization through three 20-min washes at room temperature (RT) on a rotating wheel in 1 ml of PBST × 0.2% (1× phosphate-buffered saline (PBS) with 0.2% Triton X-100). Following this, the embryos were blocked overnight at 4 °C on a rotating wheel in 1 ml of PB1T × 0.2% (1× PBS, 0.2% Triton X-100, and 1% bovine serum albumin (BSA)). Primary antibody incubation was performed for 2 h at RT for H3K27ac and H3K27me3. After washing with PBST × 0.2%, the secondary antibody (Alexa Fluor, Thermo Fisher Scientific) and DAPI (Sigma-Aldrich, D9542) were incubated for another 2 h at room temperature in PB1T × 0.2% to a final concentration of 10 μg/mL. The embryos were then washed another four times with PBS + 0.2% Triton. Finally, embryos were mounted on microscope slides (76 mm by 26 mm by 1 mm; Marienfeld) and precision coverslips (22 mm by 22 mm; Marienfeld), using VECTASHIELD Antifade Mounting Medium (H-1000-140). All images were acquired using the confocal laser scanning microscopes, Zeiss LSM 880 Airyscan. Stacks were compiled using ImageJ or Imaris 9.5.1 (Bitplane). A full list of antibodies in use is listed in Supplementary Data 8. Nuclei number are counted from different apical sections of the same area in three embryos for each condition.

## Cloning and generation of in vivo enhancer reporter lines

To generate the in vivo enhancer reporter lines to test their activity and patterning in early embryos, we adopted the same strategy from[118], with some variations. Briefly, each line contains a transcriptional reporter construct with a 2 kb enhancer DNA fragment previously tested in ref. 36, The candidate enhancer is cloned in front of the minimal DSCP promoter, which contains TATA, Inr, MTE, and DPE motifs, and the gene coding for β-galactosidase (LacZ) derived from *Escherichia coli*. The pBPGUw plasmid (Addgene, #17575), a modular Gateway GAL4 destination vector suitable for in vitro cloning and ΦC31-mediated site-specific integration in *Drosophila*, was modified by replacing the GAL4 CDS and yeast transcription terminator with the LacZ CDS using HindIII digestion (referred to as pDest-BDP-LacZw). The 2 kb enhancer fragments were amplified from genomic DNA of wild-type *Drosophila* embryos and cloned into the Gateway vector pENTR™/D-TOPO™ (#K240020, Invitrogen) according to the manufacturer's protocol to generate the so-called entry clones. The appropriate entry clones were then recombined with the pDest-BDP-LacZw using LR clonase (#11791020, Invitrogen) to generate the enhancer reporter clones. All reporters were injected and integrated on the 2nd chromosome using the attP docking site for phiC31 integrase-mediated recombination (transgene at locus 22A3). Successively, each reporter fly line has been crossed with the Nejire shRNA#1[24] or E(z) shRNA (BDSC, #33659). A full list of the fly lines in use is available in Supplementary Data 7. Maternal KD via RNAi of the respective chromatin factors is induced by mating females containing a transgene that encodes a shRNA with males carrying the Gal4-driver (BDSC, #7063) at 25 °C.

## Embryos collection and CUT&Tag

CUT&Tag experiments were performed as described in ref. 119 with small adaptations. Briefly, to enrich certain developmental stages, before cycle 9 embryos were collected for 50 min and aged at 25 °C for 40 min, cycle 10–12 embryos were collected for 1 h and aged at 25 °C for 50 min while cycle 14 embryos were collected for 1 h and aged at

25 °C for 130 min. Prior collection, egg laying of the flies was synchronized by changing plates for three consecutive times. After three pre-lays, the embryos were collected and aged 25 °C to reach the stage of interest. The plate with the embryos was removed from the incubator and the embryos were dechorionated with 50% bleach: sodium hypochlorite (6–14% active chlorine) for 2 min at room temperature. After, dechorionated embryos were collected in a 70 μm pore sieve, washed with water, dried and immediately placed in 10 mL of heptane and crosslinked with 5 mL of 1% PFA in buffer A (60 mM KCl, 15 mM NaCl, 15 mM HEPES [pH 7.6] (Sigma-Aldrich, # H0887), and 4 mM MgCl$_2$) for 15 min at room temperature using an orbital shaker. To stop the crosslinking process, 225 mM glycine was added (final concentration) followed by 5 min incubation at room temperature. The embryos were washed with buffer A containing 0.1% Triton X-100 (Sigma-Aldrich, # T8787) and staged manually on a cooling station under a microscope with transmitted light, based on their distinct morphological features. Embryos before cycle 9 were carefully selected to mainly enrich cycles 7 and 8 in order to avoid earlier stages, which are mainly constitued by a low number of nuclei.

During the early stages of development, all nuclei undergo divisions within the syncytial blastoderm, with the number of cycles reflecting the mitotic divisions. At the ninth cycle, the nuclei move to the embryo's periphery, initiating the transcription of an initial subset of 100 genes. This nuclear migration serves as an indicator of early developmental cycles. By the end of the 14 mitotic divisions, the embryo becomes fully transcriptionally active, and membranes start to invaginate between the nuclei. For a visual representation, refer to Fig. 1a and Supplementary Fig. 1a. Embryos were finally stored at −80 °C until further use. Based on the developmental stage from before cycle 9 to cycle 14, 500, 150 or 50 hand-staged embryos were used for each antibody. Embryos are pooled and nuclei from the same genotype are isolated and equally split according to the number of antibodies in use. To extract nuclei, embryos were lysed in 50 μL of Digitonin Wash Buffer (20 mM HEPES [pH 7.5] (Jena Bioscience, #CSS-511), 150 mM NaCl (Sigma Aldrich, #S6546), 0.5 mM Spermidine (Sigma Aldrich, #S0266), 5 mM sodium butyrate (Sigma Aldrich, #303410), Roche Protease Inhibitor (Sigma Aldrich, #11873580001), 0.05% Digitonin (Millipore, #300410)). After lysis, nuclei were spin for 5 min at 600 g at 4 °C and washed with 500 μL of Digitonin Wash Buffer to remove impurities. Following, 15 μL of BioMag ConcanavalinA beads (Polysciences, #86057-3) in Binding Buffer (20 mM HEPES (pH 7.5), 10 mM KCl, 1 mM CaCl$_2$, 1 mM MnCl$_2$) were added to the sample and incubated on a wheel for 15 min at room temperature. For each condition, the pool of nuclei was split and histone H3 or another antibody was used, to use the same starting material for normalisation purposes. Next, the supernatant was removed by using a magnetic stand and beads were resuspended in 150 μL of Antibody Buffer (20 mM HEPES [pH 7.5] (Jena Bioscience, #CSS-511), 150 mM NaCl (Sigma Aldrich, #S6546), 0.5 mM Spermidine (Sigma Aldrich, #S0266), 5 mM sodium butyrate (Sigma Aldrich, #303410), Roche Protease Inhibitor (Sigma Aldrich, #11873580001), 0.01% Digitonin (Millipore, #300410), 2 mM EDTA (ThermoScientific, #15575020) together with the respective antibody and incubated on a rocking platform at 4 °C overnight. The following day, samples were mildly spin down and supernatant was removed with the help of a magnetic stand. Successively, the samples were washed twice with Antibody Wash Buffer, finally resuspended in 150 μL of Antibody Buffer supplemented with the secondary antibody and incubated for 1 h at room temperature on a rocking platform. After two washes with Antibody Wash Buffer, the samples were resuspended in Digitonin Med Buffer (20 mM HEPES [pH 7.5] (Jena Bioscience, #CSS-511), 300 mM NaCl (Sigma Aldrich, #S6546), 0.5 mM Spermidine (Sigma Aldrich, #S0266), 5 mM sodium butyrate (Sigma Aldrich, #303410), Roche Protease Inhibitor (Sigma Aldrich, #11873580001), 0.01% Digitonin (Millipore, #300410)) supplemented with the pA/G Tn5 (1:150) and incubated for 1 h at room temperature on a rocking

platform. The samples were mildly spin down and two washes were performed before starting the tagmentation reaction by adding 150 μL of Tagmentation Buffer (20 mM HEPES [pH 7.5] (Jena Bioscience, #CSS-511), 300 mM NaCl (Sigma Aldrich, #S6546), 0.5 mM Spermidine (Sigma Aldrich, #S0266), 5 mM sodium butyrate (Sigma Aldrich, #303410), Roche Protease Inhibitor (Sigma Aldrich, #11873580001), 10 mM MgCl₂) and by incubating the reaction in a thermomixer for 1 h at 37 °C. The reaction was inhibited by adding 20 mM EDTA (final), 0.5% SDS (final) and 10 mg Proteinase K (final). The samples were incubated for 30 min at 55 °C and another 20 min at 70 °C. DNA purification was performed by using the ChIP DNA Clean & Concentrator kit (Zymo Research, #D5205) following manufacturer instructions. Libraries were prepared by supplementing the NEBNext HighFidelity 2× PCR Master Mix with 1 pg of Tn5-tagmented lambda DNA (New England Biolabs, #N3011S) spike-in as normalizer supplemented in the NEBNext HighFidelity 2× PCR Master Mix (New England Biolabs, #M0541S) for each sample. The indexing was performed using Illumina i5 and i7 through 15 cycles (1 × 5 min at 72 °C, 1 × 30 s at 98 °C, 13 × 10 s at 98 °C, 30 s at 63 °C, 1 × 1 min at 72 °C, hold at 4 °C). The libraries were purified using Nucleomag NGS Clean-up and Size Select beads (Macherey-Nagel, #744970.50). Quality checks were done with Qbit DNA HS Assay (ThermoScientific, #Q32854) and Bioanalyser (Agilent, #5067-4626). Finally, libraries were pooled and sent for next generation sequencing. A full list of antibodies in use is listed in Supplementary Data 8.

## Bulk ATAC-seq

ATAC-seq experiments were performed as in ref. 23 with small adaptations. Briefly, embryos were dechorionated, fixed and collected as described for CUT&Tag. To extract nuclei, embryos were lysed in 50 μL of ATAC Wash Buffer (20 mM HEPES [pH 7.5] (Jena Bioscience, #CSS-511), 50 mM NaCl (Sigma Aldrich, #S6546), 0.5 mM Spermidine (Sigma Aldrich, #S0266), 5 mM sodium butyrate (Sigma Aldrich, #303410), Roche Protease Inhibitor (Sigma Aldrich, #11873580001), 0.05% Digitonin (Millipore, #300410)). Nuclei were spin for 5 minutes at 600 g at 4 °C and washed with 500 μL of ATAC Wash Buffer to remove impurities. For each condition, the pool of nuclei was split in two: half of the nuclei were processed for CUT&Tag against histone H3 for normalisation purposes, while the other part of nuclei was treated for ATAC-seq. Tagmentation reaction was started by resuspending nuclei in ATAC Tagmentation buffer (20 mM HEPES [pH 7.5] (Jena Bioscience, #CSS-511), 50 mM NaCl (Sigma Aldrich, #S6546), 0.5 mM Spermidine (Sigma Aldrich, #S0266), 5 mM sodium butyrate (Sigma Aldrich, #303410), Roche Protease Inhibitor (Sigma Aldrich, #11873580001), 10 mM MgCl₂) supplemented with 2 μL of Tn5. The reaction was incubated in a thermomixer for 30 minutes at 37 °C. Immediately after, samples were purified using the ChIP DNA Clean & Concentrator kit (Zymo Research, #D5205) following manufacturer instructions. Libraries were prepared by supplementing the NEBNext HighFidelity 2× PCR Master Mix with 1 pg of Tn5-tagmented lambda DNA (New England Biolabs, #N3011S) spike-in as normalizer supplemented in the NEBNext HighFidelity 2× PCR Master Mix (New England Biolabs, #M0541S) for each sample. The indexing was performed using Illumina i5 and i7 (Buenrostro et al., 2015) through 15 cycles (1 × 5 min at 72 °C, 1 × 30 s at 98 °C, 13 × 10 s at 98 °C, 30 s at 63 °C, 1 × 1 min at 72 °C, hold at 4 °C). The libraries were purified using Nucleomag NGS Clean-up and Size Select beads (Macherey-Nagel, #744970.50). Quality checks were done with Qbit DNA HS Assay (ThermoScientific, #Q32854) and Bioanalyser (Agilent, #5067-4626). Finally, libraries were pooled and sent for next generation sequencing.

## Native embryos collection, nuclei isolation and 10× Multiome ATAC + Gene Expression

Single-cell multiome datasets were generated starting from 100 hand-selected cycle 14 embryos of wild-type, CBP-KD or E(z)-KD

conditions. Two biological replicates were independently prepared and processed using the Chromium Single Cell Multiome ATAC + Gene expression kit (scATAC-seq + scRNA-seq). Control or KD embryos were independently collected for 1 h on apple juice agar plates and aged at 25 °C for 140 min to enrich cycle 14 timepoint. Dechorionation was performed for 2 min with 50% "bleach" solution: sodium hypochlorite (6–14% active chlorine). Embryos were collected in a 70-μm pore size sieve and washed with distilled water. Dechorionated embryos were collected with a brush and transferred to 1.5 mL Wash Buffer (10 mM Tris-HCl [pH 7.4] (Sigma-Aldrich, # T2194), 10 mM NaCl (Sigma-Aldrich, #S6546), 3 mM MgCl₂ (Sigma-Aldrich, #M1028), 1% BSA, 0.1% Tween-20 (Sigma-Aldrich, #P9416)). Hand-staging of cycle 14 embryos is performed on a cooling station under a microscope with transmitted light according to their morphology. Nuclei isolation was performed according to the protocol provided from 10× Genomics (demonstrated protocol CG000366, Rev A) with small adaptations. Around 100 hand-selected embryos at cycle 14 are immediately transferred into 50 μL of 0.1× Lysis Buffer composed by diluting 1× Lysis Buffer (10 mM Tris-HCl [pH 7.4] (Sigma-Aldrich, # T2194), 10 mM NaCl (Sigma-Aldrich, #S6546), 3 mM MgCl₂ (Sigma-Aldrich, #M1028), 1% BSA (Sigma-Aldrich, #A9418), 0.1% Tween-20 (Sigma-Aldrich, #P9416), 0.1% NP-40 (Sigma-Aldrich, #CA630), 0.01% Digitonin (Millipore, #300410)) with Lysis Dilution Buffer (10 mM Tris-HCl [pH 7.4] (Sigma-Aldrich, #T2663), 10 mM NaCl (Sigma-Aldrich, #S6546), 3 mM MgCl₂ (Sigma-Aldrich, #M1028), 1% BSA (Sigma-Aldrich, #A9418)) and supplemented with 1U/μL RiboLock RNAse inhibitor (ThermoScientific, #EO0381). The lysis is performed in a 0.5 mL glass douncer (Kimble, #885300-0000) by applying 5 strockes. The lysate is incubated for 3 min on ice and the reaction is stopped by adding 100 μL of Wash Buffer. Successively, the nuclei suspension is transferred in a protein low-binding 1.5 mL tube containing 350 μL of Wash Buffer and immediately centrifuged at 600 g for 5 min at 4 °C. The supernatant is removed, the pellet is resuspended in 1 mL of Wash Buffer and filtered through a 10-μm pore size sieve (Cell Strainer, # 43-10010-50). The flow-through is centrifuged at 600 g for 5 min at 4 °C. Integrity and roundness of nuclei was checked by staining 10 μl of nuclei suspension with DAPI. Successively, nuclei were counted using a Trypan Blue assay (ThermoScientific, #T10282) on the automated cell counter Countess (ThermoScientific). Single-cell encapsulation and library preparation were performed according to the manufacturer's protocol (Chromium Next GEM Single Cell Multiome ATAC + Gene Expression User Guide, CG000338). Quality checks were done with Qbit DNA HS Assay (ThermoScientific, #Q32854) and Bioanalyser (Agilent, #5067-4626). Finally, libraries were pooled and sent for next generation sequencing.

## Preparation of nuclear extract and FACS sorting of mesoderm specific nuclei followed by ChIP-seq

FACS sorted mesoderm nuclei were kindly provided by Furlong lab. Briefly, they were obtained from *twi^PEMK::SBP-His2B* embryos at the 6–8 h stage of development as described in ref. 44. Immediately after sorting, chromatin immunoprecipitation is performed for H3K27me3 (Diagenode, C15410195) as described in ref. 120. Next, libraries were pooled and sent for next generation sequencing.

## Total or nuclear bulk RNA-seq

To enrich and collect exact stages of embryo development, the flies were allowed to lay eggs on apple juice agar plates in cages for one hour at 25 °C. Prior collection, egg laying of the flies was synchronized by changing plates for three consecutive times. After three pre-lays, the embryos were collected for one hours and aged 25 °C to reach the stage of interest. The plate with the embryos was removed from the incubator and submerged with halocarbon oil 27 (Sigma-Aldrich, #H8773) to allow hand-staging using a stereo-microscope.

For total RNA-seq, embryos of the correct developmental stage were then transferred into 50 μl of Trizol (ThermoScientific, #15596026), dissociated with a tissue grinder and snap-frozen and stored at −80 °C until RNA extraction. For nuclear RNA-seq, embryo collection and nuclei isolation were perfomed as described before and the nuclear pellet was resuspended in 50 μl of Trizol. Before starting with RNA-extraction, the Trizol solution was defrosted and filled up to 500 μl supplemented with 1 μL of 1:1000 ERCC RNA spike-in (Ambion, #4456740). After the addition of 100 μl chloroform and mixing, the samples were spun down at 4 °C, 12,000 × *g* for 15 min to separate the organic and aqueous phase. The aqueous phase was recovered and transferred into 300 μl of chloroform and the first step was repeated. Right after, the aqueous phase was transferred into isopropanol containing 20 μg of glycogen. The RNA was precipitated at −20 °C overnight and pelleted by centrifugation at 12,000 × *g* for 60 min, 4 °C. The pellet was washed with 80% ethanol, air dried for 5 min and resuspended in water. The RNA was quantified using Qubit RNA BR Assay kit (ThermoScientific, #Q10210) and integrity was assessed on agarose gel or through the BioAnalyser (Agilent, #5067-4626). Libraries were generated using the Illumina Stranded Total RNA-seq kit with Ribo-zero Plus (Illumina, #20072063) according to the manufacturer's instructions followed by followed NEBNext Ultra Directional RNA Library Prep Kit for Illumina (New England Biolabs, #E7420). Libraries were pooled and sent for next generation sequencing.

## cDNA synthesis and RT-qPCR

Total RNA was extracted as described before from 25 embryos at cycle 14 (ZGA). After DNAse inactivation, RNA was used for cDNA syntesis (ThermoScientific, #K1612) followed by qPCR. The mRNA levels were normalized to the ribosomal gene Rp49.

Primer pair sequences used in this study (5′ to 3′):

Zelda: GAGTTCCCCAGCACCACTAC and GCCGTGCAGTTGTACGTTTT

GAF: TCTAGTAGCGGATCCAGCGG and GTGCAGGGATTGAACCGTCT

Rp49: CTAAGCTGTCGCACAAATGG and GGGCATCAGATACTGTCCCT.

## Probe generation for in situ hybridisation

Probes targeting the gene of interest were generated by amplifying fragments from cDNA, incorporating a T7 and a T3 promoter by using respectively forward and reverse primers. Fragments of the expected size were subcloned into pJET 1.2 (ThermoScientific, #K1232) and confirmed by sequencing. To generate the probes, genes were PCR-amplified from plasmid DNA, purified, and subjected to in-vitro transcription and DIG-labeling using the Roche DIG RNA Labeling Mix (#11277073910) following the manufacturer's instructions. Following DNase digestion and clean-up, the samples underwent alkaline hydrolysis to achieve a final probe length of 500–800 bp (60 mM Na2CO3, 40 mM NaHCO3 final concentration at 60 °C). The reaction was stopped by adding 50 mM sodium acetate and 0.2% glacial acetic acid at final concentrations. After clean-up, the probes were kept at −80 °C until further use.

## In situ hybridisation

In situ hybridisation was performed as described in ref. 22. Briefly, control and KD embryos were collected separately for 0–240 min to enrich for stage 5 (cycle 14), dechorionated and transferred to 10 mL heptane before 5 mL of the fixative (4% PFA in 1× PBS) were added. Embryos were shaken for 20 min and the aqueous layer was removed. 5 mL of methanol were added and embryos were vortexed to favour devitellinization process. Embryos settled to the bottom of the tube were collected, washed three times with ice-cold methanol and

ultimately stored at −20 °C until further use. To start in situ hybridization, approximatively 200 μl of embryos were transferred into Roticlear (Carl Roth, #64742-48-9) and washed for 10 min on a rocker. Subsequently, the embryos were transferred to 100% methanol, rinsed twice with permeabilization buffer (0.05% deoxycholic acid, 7.5 mM Tris-HCl [pH 8.8], 0.05% Saponin, 0.2% BSA, 0.1% Triton-X100, 0.05% NP40), and incubated for 2 h at 4 °C in permeabilization buffer on a nutator. Next, embryos underwent fixation once more with 4% PFA for 20 min at room temperature, followed by five washes in 1 × PBS + 0.1% Triton. After the final wash, the embryos were incubated for 10 minutes on a nutator in a 1:1 mixture of 1 × PBS + 0.1% Triton and hybridization buffer (1% Roche blocking reagent #11096176001, 0.1% CHAPS, 50% formamide, 5× SSC, 1× Denhardt's solution, 0.01% Herring sperm DNA (100 μg/ml), 0.01% Heparin (100 μg/ml), 0.1% Tween-20). Before adding the probe, embryos were incubated for 1–2 h at 56 °C in hybridization buffer. The probe, denatured to prevent secondary structures at 90 °C, was snap-cooled and added at a final concentration of 2 ng/μl in hybridization buffer to the embryos that were then incubated overnight at 56 °C. On the following day, the probe was removed and the embryos were rinsed in hybridization buffer before undergoing four 15-min washes at 56 °C. Another wash in a 1:1 mixture of 1 × PBS + 0.1% Triton and hybridization buffer was performed (15 min at RT on a nutator) before washing an additional five 5-min washes in 1 × PBS + 0.1% Triton at room temperature. Successively, the embryos were blocked with 5% BSA in 1 × PBS + 0.1% Triton for 1 h on the nutator. Finally, the α-DIG antibody, diluted 1:500 with 1.5% BSA in 1 × PBS + 0.1% Triton, was added and incubated for at least two hours or overnight. Afterward, the embryos were washed five times for 10 min with in 1 × PBS + 0.1% Tween to remove residual antibody. To develop the staining, the embryos were washed three times for 5 min in AP-wash buffer (0.1 M Tris-HCl [pH 9.5], 0.1 M NaCl, 50 mM MgCl2, 1 mM Tetramisol, 0.1% Tween20). Finally, freshly prepared AP-color buffer (0.1 M Tris-HCl [pH 9.5], 0.1 M NaCl, 0.1% Tween20, 0.45 mg/ml NBT, 0.175 mg/ml BCIP) was added to the embryos. Once the staining was revealed, the reaction was stopped by washing the embryos in 1 × PBS + 0.1% Tween. Before mounting, the embryos were rinsed four times with 100% ethanol, once with 1 × PBS + 0.1% Tween, and ultimately preserved in 50% glycerol.

## Analysis

**CUT&Tag.** The CUT&Tag data processing was done as in ref. 25. Briefly, we used snakePipes version 2.4.0[121] (parameters: −trim −fastqc −properPairs −dedup −mapq 1) with specific CUT&Tag Bowtie2 alignment option (−local −very-sensitive-local −no-discordant −no-mixed -I 10 -X 700). Biological replicates were merged for downstream analysis. Since the CUT&Tag contains lambda phage spike-ins for reliable quantification of global effects, the libraries were mapped to a constructed hybrid genome of dm6 and lambda phage (NCBI GenBank ID: J02459.1). Macs2 was used to call CUT&Tag peaks using the following options: -g dm -q 0.05 −broad[122]. H3K9me3 peaks were filtered from H3K27me3 regions due to cross-reaction of the antibody in use in E(z)-KD embryos. CUT&Tag normalized signals were generated using bamCoverage from deepTools (v.2.5.7). More specifically, the relative dm6/lambda total number of reads was used as normalization factor to rescale each sample through the bamCompare option −scaleFactor. For samples with H3 CUT&Tag, the ratio between normalizations factors (H3 vs mark-specific CUT&Tag) was passed to bamCompare's option −scaleFactor to account for differences in starting material. Coverage heat maps and profiles were created using plotHeatmap or plotProfile from deepTools (v.2.5.7). To quantify the signal under the peaks, we integrated the normalized signal in a +/− 1kb around each peak. To assess reproducibility, we utilised the multiBamSummary and plotCorrelation tools from deepTools to compute the Spearman correlation between replicates.

**Chromatin dynamics and ambivalent region definition.** To build the alluvial plot to describe the dynamics of H3K27ac and H3K27me3 peaks across early development, we used easyalluvial package v0.3.2. The list of peaks present in at least one stage was used to create the alluvial plot. ChIPseeker (v1.34.1) has been used to assign peaks to genes when the distance to the TSS is smaller than 10 kb. clusterProfiler (v4.6.2) has been used to perform gene ontology analysis using the following parameters in enrichGO function: pvalueCutoff = 0.1, qvalueCutoff = 0.1. Differential tracks across conditions are generated using bamCompare with the following options: –scaleFactorsMethod SES –centerReads –smoothLength 900 –binSize 300. Source code for data normalisation can be found https://github.com/zhanyinx/atinbayeva_paper_2023. To define ambivalent regions, we first identified peaks of H3K27ac and H3K27me3 from the CUT&Tag data, independently at each stage. Next, we merged all overlapping peaks across all stages to generate a unified list of peaks. For each peak in this list, we examined its overlap with H3K27ac and H3K27me3 peaks at a given stage. Ambivalent regions for each stage were then defined as the set of peaks that overlapped with both H3K27ac and H3K27me3 peaks at that stage.

**ATAC-seq.** The ATAC-seq data processing was done as for CUT&Tag. Briefly, we used snakePipes version 2.4.0[121] (parameters: –trim –fastqc –properPairs –dedup –mapq 1) with specific ATAC-seq Bowtie2 alignment option (–local –very-sensitive-local –no-discordant –no-mixed -I 10 -X 700) was used. Biological replicates were merged for downstream analysis. Since the ATAC-seq contains lambda phage spike-ins for reliable quantification of global effects, the libraries were mapped to a constructed hybrid genome of dm6 and lambda phage (NCBI GenBank ID: J02459.1). Macs2 was used to call ATAC-seq peaks using the following options: -g dm -q 0.05 –broad[122]. ATAC-seq normalized signals were generated using bamCoverage from deepTools (v.2.5.7). More specifically, the relative dm6/lambda total number of reads was used as normalization factor to rescale each sample through the bamCompare option –scaleFactor. We used the respective H3 CUT&Tag from the same pool of nuclei to calculate the ratio between normalizations factors (H3 vs ATAC-seq) by passing to bamCompare's option –scaleFactor to account for differences in starting material. Coverage heat maps were created using plotHeatmap from deepTools (v.2.5.7). To quantify the signal under the peaks, we integrated the normalized signal in a +/− 1 kb around each peak. To assess reproducibility, we utilised the multiBamSummary and plotCorrelation tools from deepTools to compute the Spearman correlation between replicates. Source code for data normalisation can be found https://github.com/zhanyinx/atinbayeva_paper_2023.

**ChIP-seq.** ChIP-seq data were processed as CUT&Tag data with the exception that the normalisation was done using total library size through '–normalizeUsing RPKM' option in bamCoverage from deepTools (v.2.5.7). Peak calling was done using Macs2 with the same parameters as described for CUT&Tag data analysis using the input as control.

**Bulk RNA-seq.** Reads were mapped to *D. melanogaster* (build dm6) using STAR[123], with the following options: –outSJfilterReads Unique –outFilterType BySJout –outFilterMultimapNmax 1000000 –alignSJoverhangMin 6 –alignSJDBoverhangMin 2 –outFilterMismatchNoverLmax 0.04 –alignIntronMin 20 –alignIntronMax 1000000 –outSAMstrandField intronMotif –outFilterIntronMotifs RemoveNoncanonicalUnannotated –outSAMtype BAM SortedByCoordinate –seedSearchStartLmax 50 –twopassMode Basic –quantMode TranscriptomeSAM GeneCounts. Data normalisation was performed using DESeq2 package[124], outlier detection and overall sample quality were assessed using PCA. Normalized expression values (TPMs) were plotted to evaluate expression of genes of interest across experimental conditions.

**10× Multiome (scATAC-seq + scRNA-seq).** 10× Genomics single-cell Multiome sequencing was performed to simultaneously profile gene expression and chromatin accessibility in both wild-type and mutant embryos. Two biological replicates were performed for each condition. Raw sequencing data were processed by the Cell Ranger ARC v1 pipeline, using dmel6 as reference genome. Quality control and filtering were performed on RNA and ATAC data using Seurat version 4.3.0[125], scanpy version 1.9.3[126] and Signac version 1.10.0[127].

**QC, filtering and preprocessing.** 10× Multiome is composed by single-cell ATAC-seq and single-cell RNA-seq assay, simultaneously profiled from the same nuclei. Initial analysis and quality control of scRNA-seq and scATAC-seq was performed in a separate manner. For scRNA-seq, cells with less than 300 total counts, less than 200 detected genes, or more than 20% of counts originating from mitochondrial genes were discarded to retain high-quality cells. Putative doublets were identified and eliminated using doubletFinder_v3[128]. Following filtering, 22358 good quality cells were retained. Data was preprocessed applying Seurat NormalizeData() method, which consists of total counts normalization, multiplication by a scale factor (10000) and natural-log transformation. To perform dataset integration, the anchor set was defined applying FindIntegrationAnchors() function on pre-selected features (obtained with SelectIntegrationFeatures() method). IntegrateData() function was further applied to compute the integration. To perform dimensionality reduction, 2000 high variable genes (HVGs) were identified using the FindVariableFeatures() function and the PCA was computed on scaled data. UMAP visualization for the RNA assay was obtained with umap tool function from scanpy. On the other hand, for scATAC-seq, a common set of peaks across the bed files from all replicates was generated using the reduce() function from GenomicRanges package version 1.52.1[129]. Low-quality peaks within the unified set were filtered out by retaining peaks with widths between 20 and 10,000. The quality control thresholds were selected individually for each replicate to ensure dataset-specific optimization as described in Supplementary Data 9. The filtered datasets were then merged to create a unique chromatin accessibility assay. ATAC counts matrix was normalized applying term frequency inverse document frequency method implemented in Signac. Dimensionality reduction was performed applying the Latent Semantic Indexing (LSI) technique using RunSVD() function and UMAP was computed utilizing 2 to 50 components from 'lsi' reduction, as the first dimension usually correlates with sequencing depth. Combination of both single-cell gene expression and chromatin accessibility modalities was performed by Weighted Nearest Neighbour (WNN) analysis. The co-embedding was computed on the WNN graph, constructed on a weighted combination of PCA and LSI obtained with FindMultiModalNeighbors() function. Reproducibility between biological replicates was computed by using Spearma correlation coefficients. For scRNA-seq data, the average gene expression levels were computed for each replicate, and used to compute the Spearman correlations. Similarly, for scATAC-seq, chromatin accessibility matrix was aggregated across cells within each replicate, and Spearman correlations were calculated between the aggregated profiles. The resulting Spearman correlation values were then visualised through a heatmap.

**Clustering and annotation.** Cells clustering was performed on RNA data with the Leiden algorithm, setting the resolution parameter to 1.2 and resulting in the identification of 20 distinct clusters. To annotate these clusters, expression of well-known germ layers marker genes (Supplementary Data 2) curated from[76] was evaluated in each cluster. Cluster annotation was then confirmed by computing the correlation

coefficient between the top marker genes in each cluster and the provided markers using the marker_gene_overlap function from scanpy toolkit.

**Peak-to-gene expression linkage.** We performed peak-to-linkage to identify peaks whose accessibility correlates with the expression of nearby genes, thereby revealing potential regulatory elements (Supplementary Fig. 3d). To capture wild type-specific elements, peaks from bam files including only the WT condition were called using Macs2[122] and a new chromatin accessibility assay was created by quantification of the new peak set (with FeatureMatrix() and Create-ChromatinAssay() functions). Linking was performed applying Link-Peaks() function within the Signac framework in WT only cells. For each gene, this function calculates the correlation between gene expression and the accessibility of each peak, taking into account parameters such as GC content (computed using the RegionStats() function), peak accessibility, and peak length. The linked peaks were then filtered based on significance ($P$ value < 0.05) and subsequently ranked based on their correlation coefficients. Mutants ATAC assays were re-quantified with the FeatureMatrix() function using WT peak set and merged to WT assay.

**Promoters and enhancers UMAP.** To explore the contribution of promoters and enhancers to germ layer definition, UMAPs were computed separately on the basis of promoters and enhancers peaks sets, excluding yolk and undifferentiated cells clusters. The promoter peak set is generated with the most accessible TSS (+/− 500 bp) of the 2000 previously identified HVGs. The enhancer peak set consists of linked peaks to the HVGs assessed by the peak-to-gene linkage analysis within +/− 20 kb from TSS; if more than one peak per gene was found, the one with the highest correlation score was selected. To validate the contribution of enhancers to germ layer definition, UMAP was computed based on the intersection between scATAC-seq peaks set of the wild-type cycle 14 condition and the Redfly CRM list[80]. Overlapping peaks within +/− 20 kb from TSS, excluding those in promoter regions (+/− 500 bp from the TSS), were used. After subsetting the ATAC count matrix based on those peak sets, normalization and processing was performed as mentioned above, and UMAPs were visualized based on LSI dimensionality reduction.

**Correlation RNA-ATAC (promoters vs enhancers).** The correlation between gene expression and chromatin accessibility was performed both on highly variable genes and selected germ layer marker genes. To perform correlation analysis in WT, chromatin accessibility assay was used. Peaks were annotated with annotatePeak() function from ChIPseeker package version 1.36.0[130] using the T × Db object from T × Db.Dmelanogaster.UCSC.dm6.ensGene annotation package (Team BC, Maintainer BP, 2019). For each gene, peaks found within +/− 500 bp from the TSS of any annotated isoform are defined as promoter peaks. In case more than one promoter peak was found, the one with the highest accessibility in WT cells was selected. The enhancer peak was determined based on the highest peak-to-gene linkage score within +/− 20 kb from TSS, as previously explained. Spearman correlation on the average expression and average accessibility per germ layer was then computed. For some genes, we could not identify any peak associated to TSS and/or enhancer, leading to their exclusion from the analysis. To assess the significant difference between gene expression-promoter and gene expression-enhancer correlations, Mann−Whitney $U$ test was applied.

**Differential gene expression and differential accessibility upon E(z) or CBP depletion.** Differential gene expression and accessibility analysis was computed between E(z)-KD or CBP-KD and WT condition in each germ layer by scanpy.tools.rank_genes_groups() function applying Wilcoxon rank-sum testing, and P values were corrected with the Benjamini-Hochberg procedure. Genes expressed in less than 1% of cells in any condition were excluded from the analysis.

**Correlation of promoter-GEX or enhancer-GEX log2FC upon E(z)-KD of germ layer marker genes.** To evaluate whether the differential gene expression observed between E(z)-KD and WT is associated with changes in chromatin accessibility, Spearman correlation between average log2 fold change of gene expression and promoter/enhancer accessibility per germ layer was calculated.

**Pseudo-bulk ATAC-seq.** To perform germ layer-specific analysis, subset-bam function was used to include only the cells corresponding to each germ layer. bw tracks are generated using bamCoverage. Differential tracks across cell types are generated using bamCompare with the following options: –scaleFactorsMethod SES –centerReads –smoothLength 900 –binSize 300.

**Potency estimation.** To estimate differentiation potency, the signalling entropy was computed using SCENT R package and used as a proxy for pluripotency[92]. Pluripotent cells, by definition, do not favour any specific developmental pathway. From a probabilistic standpoint, this can be described as a state of uncertainty or entropy. This method provides a quantitative measure of transcriptomic heterogeneity across cells; higher homogeneity corresponds to greater entropy, indicating a more undifferentiated, pluripotent state. Conversely, lower entropy suggests more specialized, differentiated cell states with reduced potency. Starting from the single cell RNA count matrix, this method quantifies the gene expression entropy within a PPI network. Default adjacency matrix PPI network (net13Jun12) was used after converting genes to fly orthologues. Potency score was then inferred using the Correlation of Connectome And Transcriptome (CCAT) method within InferPotencyStates() function.

**Signature scoring.** Gene signature scores were calculated using scanpy.tool.score_genes() function, specifying the gene set of interest. To generate the main germ layer signatures, annotation marker genes were used, pooling the population subtypes in the case of endoderm and ectoderm (Supplementary Data 2). To compile Zelda-dependent, H2A.Z-negative and H2A.Z-positive gene signatures, the top 80 genes based on their levels of H3K27ac in before cycle 9 CUT&Tag (Fig. 4e and Supplementary Fig. 4d) were selected for each category.

**Gene regulatory network (GRN) inference.** GRN was inferred using the wild-type scRNA-seq and scATAC-seq measurements. To build the GRN, we adopted the Pando algorithm which models the relationship between the expression of target genes related to their chromatin accessibility state and to the co-expression of transcription factors[84]. The JASPAR2020 database was used as motifs to transcription factors database. The network was inferred using the function infer_grn() with 'Signac' set as peak_to_gene_method, and considering regions within +/− 20 kb from TSSs as potential regulatory elements. To construct TF modules, find_module() function with default parameters was used. For each gene in the network, we defined its primary germ layer in wild-type, as the germ layer in which the gene has the highest average expression. To evaluate the effect of E(z) and CBP depletion on gene regulation within the network, gene ID score was computed in wild type and mutant conditions. This score represents the ratio of expressing cells within the appropriate germ layer with respect to the total number of cells expressing the gene.

**Projection in the _continuum_ of _Drosophila_ embryogenesis.** To study the temporal dynamic of the data, a sci-RNA-seq and sci-ATAC-seq

dataset composed of WT embryos collected at different time points throughout *Drosophila* embryogenesis was used as reference to project our scMultiome dataset[56]. For sci-RNA-seq data, Seurat object was downloaded from the dedicated public repository hosted on the authors website (https://shendure-web.gs.washington.edu/content/members/DEAP_website/public/) and stages from 0 to 10 h were selected. PCA, UMAP projection and neighbourhood graph were computed on reference within the scanpy framework. In the case of sci-ATAC-seq, peaks matrices from timepoints 0 to 10 h were downloaded and merged. scMultiome ATAC count matrix was re-generated utilising the reference peak set[56], and consequently LSI dimensionality reduction with 100 components was performed for both sci-ATAC-seq and scMultiome ATAC data separately, removing the first dimension as it was found highly correlated with read count per cell in both datasets. Next, the neighbourhood graph was computed from the LSI dimensionality reduction. Projection of RNA and ATAC data was performed in parallel using ingest function in scanpy to map timepoint labels. PCA and LSI embeddings were used for RNA and ATAC modalities, respectively. Relative percentage of cells in each genotype per projected time point was calculated as explained above.

**Pseudotime analysis.** To perform pseudotime analysis, single-cell RNA-seq data was pre-processed with palantir algorithm[131]. Briefly, diffusion map was computed using the first 30 principal components (PCs) applying run_diffusion_maps and determine_multiscale_space functions from palantir library. The ForceAtlas2 (FA) embedding was then generated using the first 2 PCs from the multiscale diffusion space (scanpy.tools.draw_graph). Pseudotime analysis was then performed with scFates library[132]. A principal tree was generated using the simpleppt ('ppt') approach with 150 principal points, setting ppt_lambda to 0.25 and ppt_sigma to 100. Node root was manually selected on the basis of its position in the embedding. Cells were then projected onto the tree, and pseudotime value was computed as the distance from the root, applying scfates.tools.pseudotime function.

**AlphaFold structural prediction.** Structural predictions were computed by applying AlphaFold3[102] to the entire Zelda and CBP sequence, respectively, and the local structure confidence score (pLDDT) per amino acid is calculated. Low pLDDT values indicate disordered structures of the associated residues. Protein regions with potential coiled coil structure are predicted using the NPS@ algorithm[133], which scores the probability per each residue. Selected pairs of fragments of Zelda and CBP are used as input of AlphaFold3 and the resulting structures are selected based on their ability to form a complex. Their dissociation free energy is estimated with FoldX[134], after relaxing the predicted structure with Gromacs.

**Pausing index calculation.** To calculate the pausing index, we quantified the levels of RNAPII-S5P within a region spanning +/− 300 bp around the TSS of GRO-seq genes[23]. This value was then compared to RNAPII-S5P levels within the downstream gene body, defined as the region extending from 300 bp to 5000 bp downstream of the TSS. The pausing index was determined as the ratio of RNAPII-S5P enrichment in the promoter-proximal region to its enrichment in the gene body, providing a measure of transcriptional pausing at each gene.

**Reporting summary**
Further information on research design is available in the Nature Portfolio Reporting Summary linked to this article.

## Data availability
The raw data were deposited in GEO and are available under the accession numbers GSE269361, GSE269362, GSE269365, GSE269367, GSE269368, GSE289288, GSE289289. The H3K27ac and RNAPII ChIP-seq in mesoderm sorted nuclei are available in the ENA portal under

identifier ERP000560. Source data are provided as Source Data file. Source data are provided with this paper.

## Code availability
The code for the analysis is deposited in a Github repository (https://github.com/dimadatascience/scmultiome).

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

## Acknowledgements

We thank the present and past members of Iovino laboratory, in particular S. Herur, N. Antinbayeva, M. Mazina, F. Wang, S. Mahajan, Y. Miao, D. Ibarra-Morales, M. Schulte-Sasse, F. Ciabrelli, M. Joshi and A. Garcia-Prado for the daily insightful discussions and crucial reading of the manuscript; we are particularly grateful to the Furlong lab, and expecially to E.E.M. Furlong for the invaluable contribution, including a critical review of the manuscript, insightful discussions, and the generous provision of mesoderm sorted nuclei used for ChIP-seq experiments. Additionally, we are profoundly thankful to S. Procaccia for the meticulous review of the manuscript, constructive feedback on the computational analyses, and for providing the projection analysis of the ATAC timing, which significantly improved the manuscript. We thank Y. Sun for the discussion about the preliminary computational analysis; the Shendure lab, in particular D. Calderon, X. Huang and J. Shendure for the initial feedback and discussion; S. Preissl for the initial feedback; our colleagues G. Cecere, G. Pyrowolakis, A. Pane, A. Akhtar and T. Jenuwein for crucial discussion; the Bonasio laboratory for initial discussion. We thank the Bioinformatics and Sequencing facilities at the MPI-IE; and in particular W. Deboutte for initial demultiplexing of the 10× Multiome dataset; the Imaging facility and Fly facility at the MPI-IE. We thank The Bloomington *Drosophila* Stock Center (NIH P40OD018537) and the TRiP at Harvard Medical School (NIH/NIGMS R01-GM084947) for providing fly stocks used in this study; Y. Schwartz (CBP), M. Harrison (Zld), M. Erokhin and D. Chetverina (GAF), R. Paro (BRD4) for providing antibodies. F.C. is supported by the Max Planck Society and IMPRS program. M.C.R.M. is supported by the European Union's Horizon 2020 Research and Innovation Programme under the Marie Skłodowska-Curie Actions Grant (agreement 813091). Additionally M.C.R.M. and N.C-W acknowledge the support of the DFG Research training Group MeInBio [322977937/GRK2344]. E.J.S and R.B are supported by the Max Planck-von Humboldt Research Award 2020. N.I. is supported from the Max Planck Society; DFG:CRC992, Project B06; Behrens-Weise Stiftung; CIBSS - EXC 2189; Deutsche Forschungsgemeinschaft - Project ID 192904750 - CRC 992 Medical Epigenetics. This work was partially supported by the Italian Ministery of Health with "Ricerca Corrente" and "5 × 1000" funds. Also, this project has received funding from the European Research Council (ERC) under the European Union's Horizon 2020 research and innovation programme (grant agreement No.819941) ERC CoG, EpiRIME. This work was supported by the German Research Foundation (DFG) under the German Excellence Strategy (CIBSS-EXC-2189, project ID 390939984). We acknowledge the use of BioRender.com for creating illustrations included in this manuscript.

## Author contributions

Conceptualization: F.C. and N.I. Methodology: F.C., A.P., E.L., B.E., M.C.R.M., F.Z., E.J.S, N.C-W, R.B., G.T., Y.Z., and N.I. Investigation: F.C., Y.Z., and N.I. Visualization: F.C., A.P., B.E., M.C.R.M., and Y.Z. Funding acquisition: N.I. and Y.Z. Supervision: Y.Z., N.I. Writing – original draft: F.C. and N.I. Writing – review & editing: F.C., A.P., E.L., B.E., F.Z., E.J.S., N.C.W, R.B., G.T., Y.Z., and N.I.

## Funding

## Competing interests

The authors declare no competing interests.
