## [Transparent Peer Review file · Nature Communications]

Chromatin landscape at cis-regulatory elements orchestrates cell fate decisions in early embryogenesis

Corresponding Author: Dr Nicola Iovino

Version 0:

Reviewer comments:

Reviewer #1

(Remarks to the Author)

How cell identities are established after fertilization is an important question in the field of epigenetics and developmental biology. In this manuscript entitled "Chromatin landscape at cis-regulatory elements orchestrates cell fate decisions in early embryogenesis", the authors showed that the diversity of cell identity has already emerged at the ZGA stage by performing single cell omics experiments. The cell lineage-specific gene expression is correlated with chromatin accessibility at enhancers but not promoters. They investigated the roles of two evolutionary conserved chromatin modifiers, E(z) and CBP, in the ZGA-stage early specification. Genetic perturbation experiments showed that CBP is required for cell fate specification possibly through stabilizing the chromatin binding of pioneer factors, while E(z)/H3K27me3 is largely dispensable for the cell fate specification but is required for the subsequent development possibly through preventing unnecessary enhancer functions at ZGA. Most of the data and conclusions are interesting and looked reasonable, which made significant advance in the field. The reviewer feels that the following points should be elaborated before publication.

Major points

- 1) The authors claim that "the progenitors of the different embryonic cell types are already well established at this early developmental stage" by showing the distinct cluster formation at ZGA stage from UMAP analysis (Fig. 2b). However, it is unclear to me whether this clustering really emerges upon ZGA or it can emerge due to uneven localization of preexisted maternal RNA. It is ideal to provide UMAP analysis of pre-ZGA or ZGA-inhibited embryos.
- 2) The current data analyses are insufficient to claim that H3K27me3 per se protects germ layer specific genes from aberrant CBP-mediated activation or H3K27ac deposition. To claim this, the authors should define H3K27me3-positive and -negative regions in WT embryos, and show that only H3K27me3-positive regions gain H3K27ac and CBP binding in E(z)-KD embryos.
- 3) One of the main conclusions is that chromatin accessibility at promoters is unchanged in CBP-KD embryos, but the mechanistic insight is lacking. The authors discuss that RNA pol II might maintain the accessibility (line 162-164), which is interesting to be tested. Is it impossible to try RNA pol II CUT&Tag in CBP-KD embryos?

Minor comments

- 1) Assessment of reproducibility in biological replicates (scRNA-seq, scATAC-seq, Cut&Tag) should be provided.
- 2) To what extent the CBP-KD embryos delay in development? Images of the KD embryos and counting the number of nuclei are appreciated.
- 3) The authors did not perform a network analysis. Therefore, the conclusion of "stressing the pivotal role of E(z) in preserving the proper cell-specific transcriptomic networks through the regulation of enhancer regions" in line 221-222 seems to be overstated. Similarly, because the difference in scRNA-seq data between WT and E(z)-KD embryos is modest (Fig. 3b, c, 4c, d), I feel that the conclusion of "E(z) preserves the physiological diversity of germ layer precursors" in line 207 is overstated. There are more examples that should be toned down (e.g. "novel role", "novel paradigm" in the abstract, and many more in the other parts).
- 4) The finding that CBP stabilizes pioneer factor binding is intriguing and novel. What is the potential mechanism? Discussion is helpful. Relatedly, I recommend to check the available data to confirm that the levels of RNA (or ideally protein, if any data) of the pioneer factors were not decreased in CBP-KD.
- 5) The finding of line 262-264 is not actually surprising, because previous studies have shown that CBP/P300/H3K27ac is not necessarily required for chromatin accessibility in cell lines and embryos (e.g. <https://doi.org/10.1038/s41588-019-0428-5>, <https://doi.org/10.1016/j.molcel.2022.09.005>, <https://doi.org/10.1016/j.molcel.2022.01.024>,

<https://doi.org/10.1101/gr.277577.122>). The authors may instead want to add some discussion about how the current findings are novel when compared to those previous studies.

6) Fig 3b left panel: As WT, E(z)-KD, and CBP-KD plots are overlaid, it's hard to see how much WT and E(z) are close between each other. Can this be shown separately?

7) Fig 3e. Green plots appears to be actually a "S"-shape, instead of a linear. Why?

8) Questions about CUT&Tag procedure and analysis in Extended Data Fig S1. Can the signal intensity be fairly compared between different stages? Were the number of nuclei the same in all stages? Is it possible that you've gotten more signals simply because you've used more nuclei at C14 than the earlier stages?

Reviewer #2

(Remarks to the Author)

In Cardamone et al. the authors use CUT&Tag and scATAC-seq in combination with scRNA-seq to investigate the role of the epigenetic landscape in germ layer formation in *Drosophila*. This represents the first such analysis of early *Drosophila* embryos, constituting a significant contribution to the field. While the data is beautiful and makes for a very valuable resource, however, in the current form of the manuscript, we struggled to identify the major new insights obtained from this dataset and analysis.

For example, -

The authors begin by examining the chromatin landscape in early embryos, noting ambivalent regions containing both inactive and active marks. They observe that these regions are absent in a pure mesodermal cell population and conclude that the ambivalent regions do not exist and that cell-type specific chromatin landscapes must exist in the early embryo. This conclusion is not substantiated, however, because early-stage embryos are not investigated (stage 9 and 10-12). This is especially relevant because other types of bivalent domains (K4/K27me3) do exist in embryos. We also note that there is no mention of any of the literature on actual bivalent domains, which should be corrected.

The authors then show signs of cell differentiation during ZGA, prior to any morphological differentiation. While this is interesting, it is not new as the authors also mentioned themselves in their introduction (line 55, 56). They then demonstrate that this differentiation is guided primarily by enhancer accessibility rather than promoter accessibility. They call this surprising, but lots of data on this exists, even from the same authors (e.g. Reddington et al., 2020 and Chereji et al., 2019).

Finally, the authors examine the roles of E(z) and CBP in regulating early cell fate determination by performing single cell multiome analysis in knockdown embryos during ZGA. This is the mechanistic and most interesting part of the paper, and the authors claim to show (i) E(z)-dependent H3K27me3 regulation of chromatin accessibility, which prevents ectopic expression of marker genes across germ layers, and (ii) stabilization of pioneer transcription factors by the activator CBP. Indeed, the work unambiguously demonstrates the necessity of E(z) in shaping the chromatin landscape required for cell differentiation, but this has been shown before, even by the same authors (e.g. Zenk et al., 2017; Coleman and Struhl; Deluca et al. 2020). Moreover, the evidence supporting the CBP-dependent mechanism appears weak. While the authors imply that CBP stabilizes pioneer factors, this claim is largely speculative and lacks clear supporting evidence. Other explanations for the data are also possible (What are the expression levels of pioneer factors in CBP depleted embryos? How does the effect on developmental timing impact the binding of Zelda and GAF? ...) and more experiments are needed to support this conclusion.

In a specific comment to the writing, we felt that many things were not discussed or explained but mentioned in passing, and references to earlier work are often missing. Some examples -

- In the abstract, the authors mention that *Drosophila* embryo is an excellent system, yet they do not explain why (in fact, they only explain what the challenges are).
- In the results section, the authors mention that K27me3 and K27ac marks are non-overlapping, but do not address this any further.
- In the results section, related to Extended Data Figure 1, the increase in histone modifications has been seen before. There are no references mentioning this.
- Also in the results section (line 190-192) "By using scATAC-seq or combined scATAC-seq + scRNA-seq data, we confirmed that chromatin accessibility also contributes to the differences between wild-type and KD conditions as well as germ layers (Extended Data Fig. 3f). How? The text does not explain how to interpret this data.
- Entropy score is used but not explained at all (Extended Data Figure 3h).
- It is unclear what results the authors use to draw the conclusion in line 207-208. To that point, no data was presented to support this.

In conclusion, the manuscript would benefit from a better description of data and interpretation, including the relevant literature, a greater focus on significant discoveries, and additional validation of CBP's role in stabilizing pioneer transcription factors.

Specific comments on the Figures

Figure 1

1b – One can argue if this figure is necessary but since it is there: why are K27me3 and K27ac mutually exclusive before cycle 9 (see also comment above)? It is clear they should be mutually exclusive per histone but why also at this larger scale?

1c – lots of K27 at cycle 0 becomes unassigned at cycle 10-12, can the authors discuss this?

1d – K27Ac signal does not look very peaky, while the heatmap in e suggests it is. Can the authors explain this discrepancy?

Figure 2

2c - How were enhancers assigned to genes? This should be explained in the main text. If this is based on gene expression this makes all arguments circular?

2d – Why was a comparison done between Promoters and Enhancers? Is that relevant? Same question goes for Extended Data Figure 2c.

2e, f – The “most accessible promoter peak OR highest scoring linked peak” was used. This seems to make the analysis a self-fulfilling prophecy. Are some of the relationships known and can that information be used?

2g – Is the enhancer in Figure 1g the known enhancer of *mes-2* or the assumed enhancer?

2h – What about the peak immediately left of the highlighted enhancer? Its behavior is very different than the one that is highlighted.

Figure 3

3b – What do we learn from this?

3c – Is interesting but very rough, see also 3d

3g – Enhancer accessibility looks similar in all lineages? How was this enhancer chosen?

Figure 4

3c – Is ATAC dynamics shown for promoters or enhancers or both? Same question goes for Extended Data Figure 2a.

Reviewer #3

(Remarks to the Author)

Reviewer #4

(Remarks to the Author)

Cell fate determination during embryonic development is a fundamental question in developmental biology. In this study, the authors employed a multiomics approach to investigate the chromatin landscape in early *Drosophila* embryos. Using CUT&Tag, they analyzed the genome-wide distribution of the histone modifications and H3K27ac during early embryogenesis. While most regions exhibited mutually exclusive marks, they identified some regions with both modifications, termed as “ambivalent regions”. The authors further used the 10x Multiome platform to simultaneously profile chromatin accessibility and gene expression at the ZGA stage (cycle 14) in wild-type, E (z)-KD, CBP-KD embryos.

The study systematically analyzes the roles of histone modifications, H3K27me3 and H3K27ac, and their enzymes in establishing cellular identities in the early embryo. The authors also demonstrate that enhancers have a more prominent role than promoters in directing cell fate commitment. The paper is well-written, accurately describes the data, and provides valuable new insights into the mechanisms of cell fate determination in early development. I found the study useful and have only a few minor comments:

1. Based on the data presented, E(z) appears to actively maintain H3K27me3, while CBP induces H3K27ac without directly promoting chromatin opening. Could the authors provide further mechanistic insights in the Discussion on how these two factors function together? Additionally, is there any evidence that demethylases are playing an active role in this process? It would be valuable to understand what factors actively open the chromatin before ZGA.

2. Following the point above, I would be more careful to say H3K27me3 inhibits CBP binding to enhancers.

3. The authors mention regions with “ambivalent” features, but more details are needed on how these regions are defined. While the example provided in Figure 1d is clear, the description in Figure 1e remains ambiguous. It may be premature to speculate on the potential functions of these ambivalent regions without more evidence.

4. Could the authors explore the possibility of providing structural insights into how CBP stabilizes ZELDA, perhaps using AlphaFold predictions? A brief literature review did not reveal direct evidence of a CBP-ZELDA interaction, suggesting that any such interaction could be indirect. It remains unclear how CBP contributes to the stabilization of ZELDA. The recent study (<https://www.nature.com/articles/s41467-023-40485-6>) might offer useful context or parallels for this discussion.

Version 1:

Reviewer comments:

Reviewer #1

(Remarks to the Author)

The revised manuscript has added more mechanistic insights into how CBP functions in ZGA and well addressed all my comments. Congratulations!

Reviewer #2

(Remarks to the Author)

We have read the revised version of Cardamone et al. We appreciate the additional data and discussion which have alleviated some of our concerns. We support publication of this work if the remaining concerns - listed below - can be addressed.

1. The notion of cell-type specific chromatin landscapes in the early embryo is still a bit confusing to this reviewer. What is an early embryo? If before ZGA, then this is not shown. If during ZGA, this is kind of as expected as it is in accordance with the different lineages that can be identified by RNA-seq? While the authors have changed the wording related to this in lines 115-119, there is still the suggestion that pre-zygotic modifications are important for lineage specification (below) although there is no data for this modification being present in a cell-type specific manner prior to ZGA. This suggestion should be removed if no data is provided to support it.

Abstract

We found that the pre-zygotic modification H3K27me3 suppresses ectopic activation of marker genes across embryonic tissues by modulating cis regulatory elements, thereby ensuring the precise gene regulatory network in each germ layer during zygotic genome activation (ZGA).

Discussion

We found that histone modifications on chromatin precede the active transcription of crucial genes involved in cell fate determination.

2. The notion that CBP can stabilize pioneer factors is interesting but the data supporting it is not super strong. The loss of pioneer factors in CBP depleted embryos can be indirect, and alpha-fold is an interesting tool but of course merely a prediction (as pointed out by the authors). So the authors can suggest but not conclude that CPB stabilizes pioneer factors. Please change wording in abstract (lines 42-44), results (lines 471-474), Figure 5 caption, and discussion.

Related, can the authors discuss how they think the role of CBP in stabilizing pioneer factors relates to its effect on pause release?

Reviewer #3

(Remarks to the Author)

Reviewer #4

(Remarks to the Author)

The authors have addressed all previous comments and provided well-analyzed new data. I have two additional suggestions regarding CBP/ZELDA interactions (no need to reply to this reviewer):

1. Possible interaction between TAZ/KIX and Zelda. I suspect that these two regions might be more important than the author claimed potential coiled region. Are there any predictions from AF3 regarding these interactions?
2. For future studies, it would be interesting to perform domain tiling analysis of CBP to identify specific sequences that interact with Zelda and other TFs. While this analysis is beyond the scope of the current paper, it could provide valuable insights.

Point by point response

Dear Editor and Reviewers,

We sincerely appreciate the constructive and insightful feedback provided during the review process. These suggestions have been invaluable in significantly improving the quality, novelty, and focus of our manuscript. We are particularly encouraged by the reviewers' recognition of the importance of our study and its potential to advance the field.

In this revised version, we have addressed all the points raised by the reviewers. We have incorporated additional experimental data and analyses, as suggested, which have strengthened our original conclusions and provided new mechanistic insights into the processes under investigation.

Below, we summarize the major points of our manuscript:

1. We generated a novel, simultaneous single-cell transcriptomic and epigenomic (scRNA-seq + scATAC-seq) dataset of the early zygotic genome activation (ZGA) embryo in *Drosophila* (Figure 2).
2. We discovered that, at the early ZGA stage, the transcriptional and epigenetic landscapes of enhancer regions are already fully established—a phenomenon not previously observed at this developmental stage (Figure 2).
3. We explored the role of chromatin regulators and associated histone modifications during germ layer specification by depleting E(z) and H3K27me3, or CBP and H3K27ac. Our findings reveal that E(z)-mediated H3K27me3 is essential for maintaining cell identities by preventing ectopic H3K27ac and inappropriate accessibility at cell type-specific regulatory regions. Conversely, CBP acts as a master regulator of cell fate specification by activating zygotic transcriptional programs, directing cells from an undifferentiated state toward their distinct fates (Figure 3)
4. Using multiomic analysis across developmental time points, we demonstrated that CBP depletion impairs zygotic transcription without affecting chromatin accessibility (Figure 4).
5. We found that CBP depletion destabilizes pioneer factor binding to chromatin (Figure 5)
6. We provide evidence that CBP regulates RNA Polymerase II dynamics during ZGA in embryos (Figure 6).

In direct response to the reviewers' comments, we have also:

1. Generated new transcriptomics data and analysis that demonstrates that germ layer specification during early *Drosophila* development takes place at ZGA.
2. Conducted and reported additional quality control measures, including the validation of putative enhancers based on previously published datasets.
3. Further investigated the effects of E(z) and CBP depletion on enhancer activity at ZGA by generating *in vivo* reporter lines of enhancer activity and patterning for four different enhancers from our scATAC-seq dataset (two for *Ptx1* and two for *Doc2*).
4. Constructed a gene regulatory network of the wild-type ZGA embryo and analyzed how E(z) or CBP depletion disrupts cell identity in this network.
5. Provided qPCR and RNA-seq analysis of pioneer factor expression levels upon CBP depletion to rule out possible unspecific effects.
6. Generated a structure of Zelda and CBP potential binding using AlphaFold to suggest CBP's role in stabilizing pioneer factor binding on chromatin.
7. Studied RNAPII dynamics in CBP-depleted embryos using CUT&Tag for Ser5/Ser2 phosphorylation states and BRD4, which regulates elongation. Moreover, we show that RNAPII pausing could be responsible for the maintenance of chromatin accessibility upon CBP loss.

8. Incorporated additional references and revised key sections based on reviewer suggestions.
9. Developed a web interface to allow readers to interact directly with our single-cell dataset, which will be made publicly available.

We believe this is the first comprehensive study to provide mechanistic insights into how chromatin regulators and their associated histone modifications orchestrate cellular identity establishment during ZGA and early embryogenesis. Our findings suggest CBP's role in stabilizing pioneer factor binding and regulating RNAPII dynamics. Furthermore, we anticipate that our single-cell multiomic dataset will serve as a valuable and transformative resource for the field of developmental epigenetics.

Thank you again for your thoughtful and thorough reviews. We hope that our revisions and new data fully address the reviewers' concerns and enhance the overall impact of the manuscript.

Reviewer #1 (Remarks to the Author)

How cell identities are established after fertilization is an important question in the field of epigenetics and developmental biology. In this manuscript entitled "Chromatin landscape at cis-regulatory elements orchestrates cell fate decisions in early embryogenesis", the authors showed that the diversity of cell identity has already emerged at the ZGA stage by performing single cell omics experiments. The cell lineage-specific gene expression is correlated with chromatin accessibility at enhancers but not promoters. They investigated the roles of two evolutionary conserved chromatin modifiers, E(z) and CBP, in the ZGA-stage early specification. Genetic perturbation experiments showed that CBP is required for cell fate specification possibly through stabilizing the chromatin binding of pioneer factors, while E(z)/H3K27me3 is largely dispensable for the cell fate specification but is required for the subsequent development possibly through preventing unnecessary enhancer functions at ZGA. Most of the data and conclusions are interesting and looked reasonable, which made significant advance in the field. The reviewer feels that the following points should be elaborated before publication.

We sincerely thank the Reviewer for the positive feedback on our manuscript and for recognizing its potential impact on the field of epigenetics and developmental biology. In this revised version of the manuscript, we have expanded on the key points raised, incorporating additional experiments to further support our conclusions.

Major points

1) The authors claim that "the progenitors of the different embryonic cell types are already well established at this early developmental stage" by showing the distinct cluster formation at ZGA stage from UMAP analysis (Fig. 2b). However, it is unclear to me whether this clustering really emerges upon ZGA or it can emerge due to uneven localization of preexisted maternal RNA. It is ideal to provide UMAP analysis of pre-ZGA or ZGA-inhibited embryos.

We sincerely appreciate the Reviewer's comment and have addressed the concern raised. For the single-cell Multiome, we isolated nuclei (and not cells) from cycle 14 embryos (ZGA) to minimize contaminations from maternal RNAs. We are sorry for not being so clear on this point, and we have further expanded the nuclei preparation protocol details in the Methods section for the revised version of the manuscript.

To answer the Reviewer's question regarding the timing of germ layers emergence, we performed bulk RNA-seq before cycle 9 embryos (prior to zygotic transcription) and at cycle 14 (ZGA). In this analysis, we did not isolate nuclei, allowing us to include both maternal and zygotic RNAs. We then examined

the expression timing of the marker genes that we previously used for germ layer annotation. Differential expression analysis revealed that cell identity markers are actively expressed only at the time of zygotic genome activation (cycle 14) and not earlier. **This new data is now included in the revised manuscript in Figure 2i.**

To further validate our findings with an orthogonal approach, we conducted a more detailed analysis of our scRNA-seq dataset by projecting it onto the previously published developmental *continuum* of *Drosophila* embryogenesis¹. Incorporating the annotation of our projected multiome cells; we found that the cluster of cells before ZGA (0-2 hours) is predominantly enriched in yolk and undifferentiated cells. Notably, the 1-3 hours group, which corresponds to the initiation of ZGA, showed enrichment of the anterior-posterior endoderm and ventral ectoderm, consistent with our germ layer pseudotime analysis (Extended Data Figure 7f,h). Furthermore, we observed that the full establishment of cellular identities coincides with the mid-developmental time points (composed by 2-4 hrs and 3-7 hrs). Together, these new analyses further support that germ layer specification occurs at the time of ZGA. **We have included this new data in Extended Data Figure 7e.**

2) The current data analyses are insufficient to claim that H3K27me3 per se protects germ layer specific genes from aberrant CBP-mediated activation or H3K27ac deposition. To claim this, the authors should define H3K27me3-positive and -negative regions in WT embryos, and show that only H3K27me3-positive regions gain H3K27ac and CBP binding in E(z)-KD embryos.

We thank the Reviewer for the useful comment, and we have now improved our visualization of the H3K27ac gain upon E(z) depletion. We have generated a heatmap of the differential H3K27ac signal between E(z)-KD and wildtype cycle 14 embryos, ranked on the H3K27me3 enriched regions in wildtype. This heatmap clearly showed that regions which bear more H3K27me3 are also accumulating higher levels of H2K27ac upon E(z) depletion, rather than regions at the bottom of the heatmap. **We have now included this new analysis in the revised manuscript in Extended Data Figure 5g,** as the reviewer suggested.

3) One of the main conclusions is that chromatin accessibility at promoters is unchanged in CBP-KD embryos, but the mechanistic insight is lacking. The authors discuss that RNA pol II might maintain the accessibility (line 162-164), which is interesting to be tested. Is it impossible to try RNA pol II CUT&Tag in CBP-KD embryos?

We greatly appreciate the Reviewer’s insightful comment, which prompted us to delve deeper into RNA Polymerase II (RNAPII) dynamics during ZGA. To address this, we performed CUT&Tag experiments of RNAPII Serine 5 phosphorylation (RNAPII-S5P) and Serine 2 phosphorylation (RNAPII-S2P), representing the initiating and elongating forms of RNAPII, respectively.

Our analysis focused on zygotically active genes, defined using nascent GRO-seq data² and categorized based on their dependency on Zelda for transcriptional activation³. The absence of CBP caused a reduction in RNAPII-S5P loading, specifically at Zelda-dependent genes. Interestingly, this resulted in an increased pausing index in CBP-depleted embryos, suggesting a defect in the transition of RNAPII from initiation to productive elongation. This observation aligns with the global reduction in RNAPII-S2P levels, particularly along gene bodies, reinforcing the idea that CBP is required for efficient RNAPII elongation.

To further explore this mechanism, we performed CUT&Tag for BRD4, a critical factor in the elongation process⁴. Our results revealed a decreased BRD4 binding in the absence of CBP, specifically at Zelda-dependent genes. Intriguingly, the increased RNAPII pausing observed upon CBP depletion was accompanied by maintained chromatin accessibility at both enhancers and promoters of key developmental genes as recapitulated by the pseudobulk scATAC-seq, even in the absence of transcription. This finding suggests that RNAPII stalling may help preserve an open but transcriptionally inactive chromatin state in CBP-deficient embryos. **This new data has been incorporated into the revised manuscript as Figure 6 and Extended Data Figure 10.**

Minor comments

1) Assessment of reproducibility in biological replicates (scRNA-seq, scATAC-seq, Cut&Tag) should be provided.

We thank the Reviewer’s comment, and we have now performed correlation of biological replicates for all the NGS data we have provided in this manuscript. **This data has been added in the following**

figures: Extended Data Figure 1g, Extended Data Figure 3b, Extended Data Figure 4e-h, Extended Data Figure 5b,c, Extended Data Figure 8d and Extended Data Figure 10c.

2) To what extent the CBP-KD embryos delay in development? Images of the KD embryos and counting the number of nuclei are appreciated.

We thank the Reviewer for the suggestion. To answer this point, we performed DAPI staining of cycle 14 embryos across conditions. We did not observe dramatic differences in morphology nor nuclei number across conditions, except for the expected twisted phenotype and a slight reduction of nuclei upon CBP depletion⁵. However, only about 20-30% of the CBP-KD and E(z)-KD embryos reached cycle 14, and none of them completed embryogenesis (Extended Data Fig. 3d)^{6,7}. This observation goes in line with previous reports showing a complete arrest of embryogenesis at the ZGA stage in CBP-perturbed embryos across species, strengthening its conserved role in regulating early embryogenesis^{6,8-10}. **This new data is now added in the revised manuscript as Extended Data Figure 4c.**

Moreover, to investigate whether any maternal effects upon CBP or E(z) depletion could cause defects in embryogenesis before ZGA, we performed bulk RNA-seq in embryos before cycle 9 and at cycle 14. Notably, Principal component analysis (PCA) showed mild transcriptomic differences between wild-type and maternally depleted embryos before cycle 9, suggesting that the germline depletion of these factors does not drastically impair the maternal RNA load. In contrast, a marked transcriptomic difference was observed at cycle 14, confirming that the depletion of E(z) and CBP has a direct effect on the zygotic genome activation process. **This new data is now added in the revised manuscript as Extended Data Figure 4e.**

3) The authors did not perform a network analysis. Therefore, the conclusion of “stressing the pivotal role of E(z) in preserving the proper cell-specific transcriptomic networks through the regulation of enhancer regions” in line 221-222 seems to be overstated. Similarly, because the difference in scRNA-seq data between WT and E(z)-KD embryos is modest (Fig. 3b, c, 4c, d), I feel that the conclusion of “E(z) preserves the physiological diversity of germ layer precursors” in line 207 is overstated. There are more examples that should be toned down (e.g. “novel role”, “novel paradigm” in the abstract, and many more in the other parts).

We agree with the Reviewer’s point, and we have toned down some of the statements suggested in Line 271-272 of the revised manuscript. Additionally, we have now performed a gene regulatory network (GRN) analysis using Pando¹¹ to address this issue. We have used our 10x Multiome data to build the GRN underlying germ layers specification during the ZGA stage of early *Drosophila* embryogenesis. The GRN is modelled based on the relationship between the expression of target genes related to their chromatin accessibility state and to the co-expression of transcription factors. UMAP visualization of the GRN revealed cooperative regulation of cell type specification by pivotal TFs such as *tll* and *sog*, which were expressed in posterior endoderm and neural ectoderm, respectively. **This new analysis is now added in the revised manuscript as Figure 2j.**

Next, we examined the roles of E(z) and CBP in orchestrating germ layer dynamics during ZGA by re-analyzing the gene regulatory network (GRN) in the mutant backgrounds. To quantify changes, we calculated a gene identity (ID) score, defined as the ratio of cells expressing a given transcription factor (TF) within the expected germ layer to the total number of cells expressing that TF. Interestingly, E(z) depletion resulted in the ectopic expression of key TFs in inappropriate germ layers, thereby reducing the gene ID score. For example, *sog*, a key regulator of neural ectoderm formation^{12,13}, showed aberrant expression in unrelated cell types, leading to a lower gene ID score. In contrast, CBP depletion caused a complete loss of TF identity, resulting in the failure of germ layer specification altogether.

Taken together, our GRN analysis provides novel insights into the distinct roles of E(z) and CBP in regulating germ layer specification. **This new analysis has been added to the revised manuscript as Figure 3h.**

4) The finding that CBP stabilizes pioneer factor binding is intriguing and novel. What is the potential mechanism? Discussion is helpful. Relatedly, I recommend to check the available data to confirm that the levels of RNA (or ideally protein, if any data) of the pioneer factors were not decreased in CBP-KD.

We agree with the Reviewer's point, and we have now discussed this part more thoroughly in the discussion (lines 512-519). Indeed, CBP does not only possess a histone acetyltransferase (HAT), but it can also function as an integrator of combinatorial transcription factors and various transcriptional regulators¹⁴. It has been shown that its IDRs can act as a scaffold which supports the stability of the transcriptional complex by sustaining multiple interactions between TFs and promoting transcription¹⁵⁻¹⁹. To address any indirect effects on the RNA or protein levels of Zelda and GAF upon CBP depletion, we have performed RT-qPCR (Figure A below) and bulk RNA-seq (Figure B below) at cycle 14, which revealed no changes in their transcription. **We have added these new data in the revised manuscript as Extended Data Fig. 8e.**

These data match with our previous work⁶, where we observed no changes in Zelda and GAF protein levels upon CBP depletion by western blot. Immunofluorescence staining also confirmed that the nuclear import of these two pioneer factors is maintained in the absence of CBP⁶.

Additionally, to further dig into a potential interaction between CBP and Zelda that could explain the role of CBP in the stabilization of Zelda, we have performed AlphaFold predictions, as suggested by Reviewer #4. Intriguingly, two potential IDR regions in CBP and Zelda have the capability of forming coiled coils domains and could possibly interact, resulting in the stabilization of Zelda binding on

chromatin. We have added this new analysis in the revised manuscript as Figure 5b,c and Extended Data Figure 9c.

5) The finding of line 262-264 is not actually surprising, because previous studies have shown that CBP/P300/H3K27ac is not necessarily required for chromatin accessibility in cell lines and embryos (e.g. <https://doi.org/10.1038/s41588-019-0428-5>, <https://doi.org/10.1016/j.molcel.2022.09.005>, <https://doi.org/10.1016/j.molcel.2022.01.024>, <https://doi.org/10.1101/gr.277577.122>). The authors may instead want to add some discussion about how the current findings are novel when compared to those previous studies.

We thank the Reviewer for their insightful comment and for highlighting these important studies. In agreement with the prior reports cited, our data confirm that CBP and H3K27ac play a relatively minor role in regulating chromatin accessibility during cycle 14 (ZGA) in *Drosophila* embryos. However, the specific factors responsible for maintaining chromatin accessibility in the absence of CBP have not been fully characterized before.

To address this, we investigated the binding dynamics of pioneer factors, key regulators of chromatin accessibility²⁰⁻²², following CBP depletion. Our results show that CBP depletion destabilizes pioneer factor binding. To further explore this, we used AlphaFold to predict a potential interaction between CBP and Zelda, which may help explain CBP's role in stabilizing chromatin accessibility (see point #4 above, new Figure 5b, c, and Extended Data Figure 9c). Interestingly, we also observed increased pausing of RNAPII at both enhancers and promoters, suggesting that paused RNAPII may contribute to maintaining chromatin in an accessible but transcriptionally inactive state, which has never been shown before (new data added in Figure 6 and Extended Data Figure 10 of the revised manuscript).

While a recent study²³ reported chromatin compaction upon CBP inhibition in mouse embryonic stem cells (mESCs), the effects were most pronounced 30 minutes after treatment, suggesting that temporal dynamics play a critical role in this process.

Taken together, our findings provide new insights into how chromatin accessibility is maintained in the absence of CBP, emphasizing the interplay between pioneer factors, RNAPII pausing, and chromatin accessibility. In response to the Reviewer's comment, we have added the suggested references to prior studies in the revised manuscript and further elaborated on these works to clarify the novelty of our

findings. These revisions are highlighted in green in Lines 462-466 and 521-527 of the revised manuscript.

6) Fig 3b left panel: As WT, E(z)-KD, and CBP-KD plots are overlaid, it's hard to see how much WT and E(z) are close between each other. Can this be shown separately?

We agree with the Reviewer's point, we have now generated UMAP of scRNA-seq + scATAC-seq split by condition, to help the readers discerning differences between wild-type and chromatin factors depleted cells. **We have included this new data in Extended Data Figure 5b.**

7) Fig 3e. Green plots appears to be actually a "S"-shape, instead of a linear. Why?

We thank the reviewer for raising this observation. In Figure 3e, the green plot indeed deviates from a strict linear trend, forming what may appear as an "S"-shape. This phenomenon is primarily due to the limited number of genes at the extremes of very low or very high changes in expression and accessibility. To statistically improve the stability of our analysis, we binned the data and plotted the averages with their standard deviations. However, when the gene count is low in these extreme regions, the estimates become noisier, leading to deviations from linearity. These deviations are thus not indicative of a true non-linear relationship but rather reflect statistical noise inherent to sparsely populated bins.

8) Questions about CUT&Tag procedure and analysis in Extended Data Fig S1. Can the signal intensity be fairly compared between different stages? Were the number of nuclei the same in all stages? Is it possible that you've gotten more signals simply because you've used more nuclei at C14 than the earlier stages?

We thank the Reviewer for raising this important point and apologize for not being clearer in our initial submission. To be able to compare different stages, we performed CUT&Tag for total H3 and used the signal as a control to normalize for the number of nuclei at each stage. This approach assumes that total H3 levels accurately reflect the starting material, as also done for ChIP-seq²⁴ and illustrated in the scheme provided in Supplementary Figure 1b. Additionally, we incorporated a spike-in to ensure quantitative accuracy in our measurements. This methodology is now explained in detail in both the main text (line 98-101) and the Methods section of the revised manuscript.

Reviewer #2 (Remarks to the Author)

In Cardamone et al. the authors use CUT&Tag and scATAC-seq in combination with scRNA-seq to investigate the role of the epigenetic landscape in germ layer formation in *Drosophila*. This represents the first such analysis of early *Drosophila* embryos, constituting a significant contribution to the field. While the data is beautiful and makes for a very valuable resource, however, in the current form of the manuscript, we struggled to identify the major new insights obtained from this dataset and analysis.

We sincerely thank the Reviewer for their thoughtful comments and careful evaluation of our manuscript. In the revised version, we have incorporated additional experiments, validations, and analyses to address the Reviewer's points and provide deeper insights into the study. We believe that these new data and refinements have significantly strengthened the manuscript, and we hope that the Reviewer will share our excitement about the new insights and clarity presented in this revised version.

For example,

The authors begin by examining the chromatin landscape in early embryos, noting ambivalent regions containing both inactive and active marks. They observe that these regions are absent in a pure

mesodermal cell population and conclude that the ambivalent regions do not exist and that cell-type specific chromatin landscapes must exist in the early embryo. This conclusion is not substantiated, however, because early-stage embryos are not investigated (stage 9 and 10-12). This is especially relevant because other types of bivalent domains (K4/K27me3) do exist in embryos. We also note that there is no mention of any of the literature on actual bivalent domains, which should be corrected.

We thank the Reviewer for their insightful comment and fully understand their perspective. It is indeed correct that we speculated, based on mesoderm ChIP-seq data for H3K27me3 and H3K27ac, that these two modifications are differentially enriched across different cell types. However, we have not performed the same experiment in other cell types due to the lack of available markers for FACS purification during the early stages of *Drosophila* development. In light of this limitation, we have revised and toned down this section of the text (Line 115-119).

Notably, a recently published study²⁵ employed *Drosophila* genetics of Toll receptors to induce mesoderm, dorsal ectoderm, or neural ectoderm formation in embryos and performed chromatin profiling for H3K27me3 and H3K27ac in these induced lineages. Their findings revealed no overlap between the two modifications, which aligns with our interpretation. We have cited this study in the revised manuscript.

Regarding the existence of bivalent domains marked by H3K4me3/H3K27me3, while these are well-documented in vertebrates and are essential in poising genes for later activation in development²⁶⁻³², evidence in *Drosophila* remains limited. Our literature review found no direct reports of such domains in *Drosophila*³³. However, domains marked by H3K4me1/H3K27me3 have been described^{33,34}. We have clarified these distinctions in the revised manuscript (Line 112-115) and hope that the Reviewer will find this additional context helpful.

The authors then show signs of cell differentiation during ZGA, prior to any morphological differentiation. While this is interesting, it is not new as the authors also mentioned themselves in their introduction (line 55, 56). They then demonstrate that this differentiation is guided primarily by enhancer accessibility rather than promoter accessibility. They call this surprising, but lots of data on this exists, even from the same authors (e.g. Reddington et al., 2020 and Chereji et al., 2019).

We thank the Reviewer for their thoughtful comment, and we have clarified this point in the revised manuscript. While the reports mentioned by the Reviewer demonstrate that enhancer regions can exhibit differential accessibility across cell types, the data in those studies were generated at later stages of embryogenesis and not during the early embryonic ZGA stage analyzed in our manuscript. Significant effort has been dedicated to investigating chromatin heterogeneity during early development. For example, previous studies attempted to assess anterior-posterior chromatin accessibility by bisecting whole embryos^{35,36} and performing bulk ATAC-seq. While this approach provided insight into chromatin landscapes along the anterior-posterior axis, it did not resolve chromatin accessibility at the single-cell or cell-type level. Similarly, chromatin accessibility in DV tissue mutant embryos, composed of presumptive dorsal ectoderm, neurogenic ectoderm, or mesoderm²⁵, is limited in resolution due to contamination from yolk or other germ layers, as revealed by single-cell RNA-seq analyses.

In our study, we provide novel evidence that cell fate specification can be driven by enhancers as early as cycle 14 (ZGA). Using single-cell multiomic resolution from meticulously hand-selected embryos, we directly link the accessibility of thousands of cell-type-specific enhancers to the cell-type-specific expression of their target genes, despite the uniform accessibility observed at all promoter regions. Crucially, our approach simultaneously captures chromatin accessibility and transcriptomic data from the same nucleus, eliminating the need for computational integration of data from separate single-cell ATAC-seq and RNA-seq experiments, which often risks losing biological significance.

Overall, we believe our single-cell multiomic dataset from hand-sorted wild-type cycle 14 embryos provides clear evidence that enhancer accessibility, rather than promoter accessibility, is cell-type-

specific and directs target gene expression in a cell-type-specific manner as early as ZGA. This represents a significant advance in understanding how chromatin landscapes establish cell fates during the earliest stages of development and had never been proven before.

Finally, the authors examine the roles of E(z) and CBP in regulating early cell fate determination by performing single cell multiome analysis in knockdown embryos during ZGA. This is the mechanistic and most interesting part of the paper, and the authors claim to show (i) E(z)-dependent H3K27me3 regulation of chromatin accessibility, which prevents ectopic expression of marker genes across germ layers, and (ii) stabilization of pioneer transcription factors by the activator CBP. Indeed, the work unambiguously demonstrates the necessity of E(z) in shaping the chromatin landscape required for cell differentiation, but this has been shown before, even by the same authors (e.g. Zenk et al., 2017; Coleman and Struhl; Deluca et al. 2020). Moreover, the evidence supporting the CBP-dependent mechanism appears weak. While the authors imply that CBP stabilizes pioneer factors, this claim is largely speculative and lacks clear supporting evidence. Other explanations for the data are also possible (What are the expression levels of pioneer factors in CBP depleted embryos? How does the effect on developmental timing impact the binding of Zelda and GAF? ...) and more experiments are needed to support this conclusion.

We thank the Reviewer for the helpful comment. We are grateful that the Reviewer appreciated the mechanistic part about E(z)-dependent H3K27me3 role in safeguarding cell fate identities. H3K27me3 plays a role in avoiding the aberrant accumulation of H3K27ac, and this mechanism is very well conserved across species^{7,31,37-39} as the Reviewer rightly pointed. Our manuscript additionally reports that H3K27me3 prevents the aberrant opening of regulatory regions, which can induce ectopic gene expression across various cell types with the consequent loss of cellular identities. To further support this point, we have inferred a gene regulatory network by using scRNA-seq and scATAC-seq information from our 10x Multiome dataset. We observed spurious activation of germ layers-specific TFs across cell types upon E(z) depletion, resulting in a dramatic loss of cellular identities. **This new analysis has been added to the revised version of the manuscript in Figure 3h**, and we hope the Reviewer will find it convincing.

To address the Reviewer's comment regarding the CBP's role in stabilizing pioneer factors binding on chromatin, we have investigated the RNA levels of Zelda and GAF upon CBP depletion through RT-qPCR (Figure A below) and bulk RNA-seq (Figure B below) at cycle 14. This data revealed no significant changes in their transcription and **has been added in the revised version of the manuscript as Extended Data Figure 8e**. This new data matches with our previous work⁶, where we also observed no protein levels changes by western blot upon CBP-KD. Immunofluorescence staining confirmed that both Zelda and GAF are correctly localized to the nucleus in the absence of CBP⁶.

Moreover, to prove a potential mechanism by which CBP stabilizes pioneer factors and also following the Reviewer #4 suggestion, we have performed AlphaFold predictions of a potential binding between

CBP and Zelda. Our prediction does find IDR regions with coiled coil potential which can interact among CBP and Zelda. Moreover, it has been reported that CBP does not only possess histone acetyltransferase ability (HAT), but it can also function as an integrator of combinatorial transcription factors and various transcriptional regulators¹⁴. Additionally, recent reports described CBP's ability to co-condensate with different transcription factors through the interaction between different disordered regions, highlighting CBP's scaffold role in supporting the stability of the transcriptional complex, by mediating multiple TFs and co-factors interactions to promote active transcription¹⁵⁻¹⁹.

Interestingly, this phenomenon has been described in *Drosophila* embryos too, suggesting that cluster formation might also be due to Zelda and CBP colocalization at transcriptional condensates⁴⁰. These findings expand our previous *in vitro* reporter transactivation assays, in which we tested the transactivation capacity of different CBP constructs lacking different domains. Notably, the lack of the N-terminal domain of CBP, which contains the IDR interacting with Zelda predicted by our analysis, a TAZ zinc finger domain, and a KIX domain, resulted in a marked 94% decrease in CBP transactivation power, strengthening our hypothesis that CBP function in the embryo might be related with the stabilization of TFs rather than its HAT activity⁶. Nevertheless, we cannot exclude that the disruption of such a portion could also result in unstable proteins, which could be the cause of reduced transactivation. Future experiments will shed light on which concrete regions of CBP act as stabilizers of TFs binding on chromatin. Taken together, we hypothesize that CBP can mediate the stabilization of other transcription factors on chromatin and promote successful zygotic genome activation. **This new data is now included in the revised version of the manuscript in Figure 5b,c and Extended Data Figure 8c.**

In a specific comment to the writing, we felt that many things were not discussed or explained but mentioned in passing, and references to earlier work are often missing. Some examples – We sincerely apologize for the shortcomings highlighted by the Reviewer and are truly grateful for bringing these issues to our attention. In the revised version of the manuscript, we have discussed more extensively parts in the main text which were poorly explained, and we have extensively worked on the literature. We hope the Reviewer will find this version improved.

- In the abstract, the authors mention that *Drosophila* embryo is an excellent system, yet they do not explain why (in fact, they only explain what the challenges are).

We are sorry for the lack of specificity. Because of the journal's guidelines, we could not write more about this part in the abstract. We have now cited a dedicated review article⁴¹ to clarify which are the pros and cons of the *Drosophila* embryo as a model to study the relationship between chromatin dynamics and cell fate.

- In the results section, the authors mention that K27me3 and K27ac marks are non-overlapping, but do not address this any further.

We apologize for not having provided more information about this section of the results. In the revised version of the manuscript, we have included the full list of peaks from our CUT&Tags for H3K27me3 and H3K27ac throughout early *Drosophila* embryogenesis, annotated based on their dependency on H3K27me3, H3K27ac or H3K27me3/ac. This information is now available in **Supplementary Table 5**.

- In the results section, related to Extended Data Figure 1, the increase in histone modifications has been seen before. There are no references mentioning this.

We apologize for the mistake and have now added the correct reference⁴² in the revised version of the manuscript.

- Also in the results section (line 190-192) "By using scATAC-seq or combined scATAC-seq + scRNA-seq data, we confirmed that chromatin accessibility also contributes to the differences between wild-type and KD conditions as well as germ layers (Extended Data Fig. 3f). How? The text does not explain how to interpret this data.

We have now clarified this in the main text in line 234-238. By using scATAC-seq or combined scATAC-seq + scRNA-seq data, we confirmed that chromatin accessibility also contributes to the differences between wild-type and KD conditions as well as germ layers as highlighted by very well-defined clusters in both UMAP embeddings (Extended Data Fig. 5a).

- Entropy score is used but not explained at all (Extended Data Figure 3h).

We apologize that this metric was not clear enough in our previous version and have now increased the description in both Results and Method section of the revised manuscript (line 248-252). To understand if CBP-KD cells were displaying an increased tendency toward pluripotency, we used the entropy score analysis^{43,44}. Pluripotent cells, by definition, do not favor any specific developmental pathway^{45,46}. From a probabilistic standpoint, this can be described as a state of uncertainty or entropy. Using a protein-protein interaction network (PPI), we quantify the signaling promiscuity of biological processes within the network of each single condition, therefore measuring the number of active biological processes in the cell. Since a committed or differentiated cell preferentially activates specific pathways, this would manifest itself as a lower entropy rate. Our analysis revealed a higher entropy score in the CBP-KD condition, largely due to the high pluripotency in undifferentiated clusters, underscoring the essential function of CBP in promoting pluripotency exit during embryogenesis, now in Extended Data Fig. 5e).

- It is unclear what results the authors use to draw the conclusion in line 207-208. To that point, no data was presented to support this.

We agree with the Reviewer's suggestion and have now moved the conclusion to the end of the paragraph on Line 303-307 following the addition of extra data and analyses.

To summarize, our single-cell Multiome analysis, combined with data integration across conditions, revealed a redistribution of cell fractions across germ layers upon E(z) and H3K27me3 depletion. In contrast, CBP depletion led to a globally undifferentiated state, as shown previously (from previous Figure 3c).

This was caused mainly by the loss of E(z)-mediated H3K27me3, which led marker genes to be ectopically expressed across germ layers, while CBP depletion resulted in a transcriptional shutdown of these genes, as observed by both scRNA-seq and *in situ* hybridization for the RNA of two marker genes of the posterior endoderm (*Ptx1*) and dorsal ectoderm (*Doc2*). Data were presented in the precedent version of the manuscript in Figure 3f and Extended Data Figure 3j.

Moreover, by using scRNA-seq and scATAC-seq information from our 10x Multiome dataset, we have inferred a gene regulatory network (GRN) to understand the impact of chromatin factors depletion on cell fate specification during ZGA (see points above and **new Figure 3h**).

Taken together, our results suggest that H3K27me3 acts as a keeper of the proper cell state, which is lost when essential genes for cell fate are activated ectopically across germ layers. Instead, upon CBP depletion, transcription of marker genes is lost, and cells are stuck in an undifferentiated state, suggesting that CBP is pivotal in activating the transcription of marker genes and in driving cell type specification.

In conclusion, the manuscript would benefit from a better description of data and interpretation, including the relevant literature, a greater focus on significant discoveries, and additional validation of CBP's role in stabilizing pioneer transcription factors.

We thank the Reviewer for all the insightful comments and suggestions, we hope that the additional experiments analyses and literature references added in the revised version of the manuscript will clarify these points.

Specific comments on the Figures

Figure 1

1b – One can argue if this figure is necessary but since it is there: why are K27me3 and K27ac mutually exclusive before cycle 9 (see also comment above)? It is clear they should be mutually exclusive per histone but why also at this larger scale?

We agree with the Reviewer, indeed it is an interesting point we are trying to investigate further in the lab. But at the moment, we also have no answer for it.

1c – lots of K27 at cycle 0 becomes unassigned at cycle 10-12, can the authors discuss this?

The observed transition, where the H3K27 signal at cycle 9 appears unassigned by cycles 10–12, results from a lack of further signal enrichment during progression, unlike other regions. This reduced enrichment causes the signal to no longer meet the threshold for peak calling by MACS2.

1d – K27Ac signal does not look very peaky, while the heatmap in e suggests it is. Can the authors explain this discrepancy?

This is due to the different scales in both plots.

Figure 2

2c - How were enhancers assigned to genes? This should be explained in the main text. If this is based on gene expression this makes all arguments circular?

We thank the Reviewer for raising this important point and have addressed this issue. Indeed, we use gene expression to link putative enhancers to their target genes, given that we have gene expression and accessibility in the same nuclei. We have set a range of +/- 20kb from the most accessible promoter of highly variable genes in our single-cell dataset, finding that enhancers rather than promoters are better indicators of cell type specification. We have now expanded this information in the main text (line 148-153) and **added a scheme in Extended Data Figure 3d** to clarify the rationale of our analysis.

Additionally, to address the Reviewer's concern, we have overlapped our set of linked enhancers with a curated collection of known *cis*-regulatory regions⁴⁷. Interestingly, 49% of our set of linked peaks are indeed overlapping with previously identified regulatory elements, proving that this method is highly reliable. The remaining elements are possible novel regulators of cell type specific genes, which we are looking forward to investigating in the future and **now provide in Supplementary Table 6** to make this resource more useful for the community. To prove the robustness of our dataset with an orthogonal approach, we have intersected the whole ensemble of wildtype scATAC-seq peaks from Extended Data Figure 2a, excluding promoters, with the curated collection of CREs⁴⁷ and performed UMAP dimensional reduction. Again, our analysis reveals a much clearer clustering of the different germ layers when already-known enhancers are used, suggesting that enhancers are better indicators of cell type-specific chromatin states, rather than promoters. **This data has been added as Extended Data Figure 3f in our revised manuscript.**

2d – Why was a comparison done between Promoters and Enhancers? Is that relevant? Same question goes for Extended Data Figure 2c.

We aimed to investigate the roles of promoters and enhancers in cell fate specification, as these genomic elements play key roles in transcriptional regulation⁴⁸. We were particularly intrigued by the observation that promoters remained accessible and loaded with RNAPII at marker genes, even in tissues where these genes are not expressed (Fig. 2g-h and Extended Data Fig. 3h). This prompted us to explore this phenomenon further and assess the contribution of enhancers, as they may serve as stronger predictors of gene expression and play more prominent regulatory roles at this developmental stage.

2e, f – The “most accessible promoter peak OR highest scoring linked peak” was used. This seems to make the analysis a self-fulfilling prophecy. Are some of the relationships known and can that information be used?

As stated in comment 2c, we compared this set of linked peaks with a curated collection of known cis-regulatory regions⁴⁷. Interestingly, 49% of our putative enhancers overlap with previously identified regulatory elements. This demonstrates that our method is highly reliable and independent of our specific approach to selecting enhancer regions, as similar results were obtained when using previously annotated enhancer datasets.

2g – Is the enhancer in Figure 1g the known enhancer of *mes-2* or the assumed enhancer?

This enhancer is the highest-scoring linked peak in our dataset and has been previously annotated as CRM5130. It was characterized as an element with enhancer activity for *Mes2* in oenocytes and somatic muscle at stage 11 embryos⁴⁹. Additionally, it has demonstrated enhancer activity in FACS-sorted mesodermal cells from 6–8 hours embryos⁵⁰. These observations further highlight the robustness and reliability of our method.

2h – What about the peak immediately left of the highlighted enhancer? Its behavior is very different than the one that is highlighted.

The side peak shows a similar trend to the highlighted one; it is more accessible in the anterior endoderm but also accessible in other clusters. We speculate that also this region could contribute to modulating the expression of *Lim1*, or it might have a role in other cell types, while the highlighted enhancer that we find with the highest score is more likely to direct the expression of this gene in the anterior endoderm cluster.

Figure 3

3b – What do we learn from this?

We observed distinct, mutant-specific responses upon depletion of chromatin factors. Following E(z) depletion, transcription is only mildly altered compared to wild-type nuclei. While most germ layers are still formed, the proportions of individual cell types within the embryo are disrupted.

In contrast, CBP depletion completely prevents cell fate establishment, with all cells showing transcriptional profiles distinct from the control. This suggests that CBP is essential for activating zygotic transcription. Additionally, we have quantified cell proportions across germ layers under all three conditions, as shown in Figure 3c of the previous manuscript version. In response to the Reviewer’s comment, we have now expanded and clarified this analysis in the text at Line 240-246.

3c – Is interesting but very rough, see also 3d

We apologize for the lack of clarity; we have added the raw cell number data and the relative cell fractions for each condition in **Supplementary Table 8**. Additionally, we have also provided in **Supplementary Table 9** the raw counts of GEX used to generate the heat map from Figure 3d of the previous version of the manuscript.

3g – Enhancer accessibility looks similar in all lineages? How was this enhancer chosen?

We selected this enhancer peak because it overlaps with a region previously identified as actively regulating *Doc2* expression in the dorsal ectoderm⁵¹. In the revised manuscript, we generated transgenic fly lines to test the *in vivo* expression patterns and activity of enhancers identified both in our scATAC-seq peaks and in a the curated database RedFly⁴⁶. Specifically, we screened two regulatory regions for *Ptx1* and two for *Doc2* in control embryos and embryos depleted of either E(z) or CBP during ZGA. **This new data is now added in the revised manuscript as Extended Data Fig. 7.**

To assess enhancer activity, we used *in situ* hybridization staining of LacZ as a reporter for enhancer-driven expression. In E(z)-depleted embryos, we observed aberrant expression of the reporter compared to controls, supporting our hypothesis that H3K27me3 restricts enhancer activity to the appropriate cell types.

In contrast, CBP-depleted embryos showed complete disruption of the reporter expression pattern, consistent with the gene expression results for *Ptx1* and *Doc2*. This confirms the essential role of CBP as an activator of developmental enhancers. These findings have been incorporated into Extended Data Figure 7a,b.

Figure 4

3c – Is ATAC dynamics shown for promoters or enhancers or both? Same question goes for Extended Data Figure 2a

We thank the Reviewer for this comment. We previously showed the dynamics genome-wide to assess general changes. In response to the reviewer, we have now separately analyzed the percentage of cells across time points in each condition by using promoter or enhancer accessibility. For promoter accessibility, we used each peak from our dataset, which coincides with +/- 500 bp from TSS, while for enhancers, we used each peak falling in the range of +/- 20 kb from TSS. The data confirm no major differences in the temporal dynamics of chromatin accessibility across conditions, as also seen overall, and no differences in the progression between promoters or enhancers.

Reviewer #3 (Remarks to the Author):

Reviewer #4 (Remarks to the Author)

Cell fate determination during embryonic development is a fundamental question in developmental biology. In this study, the authors employed a multiomics approach to investigate the chromatin landscape in early Drosophila embryos. Using CUT&Tag, they analyzed the genome-wide distribution of the histone modifications and H3K27ac during early embryogenesis. While most regions exhibited mutually exclusive marks, they identified some regions with both modifications, termed as "ambivalent regions". The authors further used the 10x Multiome platform to simultaneously profile chromatin accessibility and gene expression at the ZGA stage (cycle 14) in wild-type, E (z)-KD, CBP-KD embryos.

The study systematically analyzes the roles of histone modifications, H3K27me3 and H3K27ac, and their enzymes in establishing cellular identities in the early embryo. The authors also demonstrate that enhancers have a more prominent role than promoters in directing cell fate commitment. The paper is well-written, accurately describes the data, and provides valuable new insights into the mechanisms of cell fate determination in early development. I found the study useful and have only a few minor comments:

We thank the Reviewer for the thoughtful and insightful feedback on our manuscript, and we feel encouraged by the recognition of our work.

1. Based on the data presented, E(z) appears to actively maintain H3K27me3, while CBP induces H3K27ac without directly promoting chromatin opening. Could the authors provide further mechanistic insights in the Discussion on how these two factors function together? Additionally, is there any evidence that demethylases are playing an active role in this process? It would be valuable to understand what factors actively open the chromatin before ZGA.

We thank the Reviewer for their suggestion, and we have expanded the mechanistic crosstalk between H3K27me3 and H3K27ac in the Discussion (Line 537-548). Briefly, based on our new data and previously published report from our lab⁷, we hypothesize that the maternally inherited deposition of H3K27me3 is responsible for promoting an inactive/repressed default genomic state during the early stages of *Drosophila* development. Kickstart of ZGA might be triggered due to the binding of remodelers and pioneer factors, which successively recruit CBP and the H3K27-specific demethylase UTX, which has been described to interact with the chromatin-remodeler Brahma (BRM) and CBP itself⁶². We speculate a possible mechanism by which the combinatorial action of those factors, coupled with other TFs, might erase the repressive barrier maternally installed at Polycomb loci and favors nucleosome remodeling in a cell type-specific fashion to prepare the correct stage for active transcription.

2. Following the point above, I would be more careful to say H3K27me3 inhibits CBP binding to enhancers.

We thank the Reviewer for pointing out this concern. We have revised the sentence in Line 269-274 to clarify that H3K27me3 influences enhancer activity, potentially by affecting CBP binding rather than directly inhibiting it.

3. The authors mention regions with "ambivalent" features, but more details are needed on how these regions are defined. While the example provided in Figure 1d is clear, the description in Figure 1e remains ambiguous. It may be premature to speculate on the potential functions of these ambivalent regions without more evidence.

We apologize for the lack of details regarding the approach used to define ambivalent regions. We have expanded this explanation in the Methods section. Briefly, peaks were called using MACS2 on H3K27ac and H3K27me3 CUT&Tag datasets. For each stage, regions showing overlap between H3K27ac and H3K27me3 signals were defined as ambivalent.

We have now provided the full set of annotated peaks, categorized based on their dependency on H3K27me3, H3K27ac, or both (H3K27me3/ac), and further divided by time point. **This new dataset has been included in Supplementary Table 5.**

Regarding the description of Figure 1e, we have clarified this in the corresponding legend, and we hope the Reviewer will find it clearer now.

4. Could the authors explore the possibility of providing structural insights into how CBP stabilizes ZELDA, perhaps using AlphaFold predictions? A brief literature review did not reveal direct evidence of a CBP-ZELDA interaction, suggesting that any such interaction could be indirect. It remains unclear how CBP contributes to the stabilization of ZELDA. The recent study (<https://www.nature.com/articles/s41467-023-40485-6>) might offer useful context or parallels for this discussion.

We thank the Reviewer for the insightful suggestion. Indeed, we did not find any report in the literature about a direct CBP-Zelda interaction, as the Reviewer pointed out. Intriguingly, these two proteins seem

to colocalize in discrete clusters immediately before the initiation of transcription, as recapitulated by live imaging⁴⁰.

To investigate the possible mechanism by which CBP stabilizes Zelda, we took advantage of AlphaFold3⁵³, as the Reviewer suggested, to model both structures and possible interaction events during transcriptional activation. To do so, a locus bearing CBP and Zelda binding by CUT&Tag in wild-type cycle 14 embryos was selected, which was also enriched for a Zelda DNA binding motif. From the structural point of view, both Zelda and CBP are predicted to display a large amount of intrinsically disordered regions (IDRs) with some structured domain. Of them, amino acid residues located in the region 696-726 of CBP and 1012-1048 of Zelda are predicted by the NPS@ algorithm to have a high propensity of forming coiled coil, a widespread protein motif that causes protein-protein interaction.

Alpafold3 confirms that these two regions bind together, causing the zinc-fingers of Zelda (residues 1326 to 1435) and of CBP (residues 508-594) to constrain their maximum distance to 144 nm, with consequent binding of Zelda through the zinc finger at a promoter region of *tll*, enriched for Zelda binding motif (chr3R:30852022-30852038), which bears both Zelda and CBP by CUT&Tag. **These new data have been included in Figure 5b,c and Extended Data Figure 9c in the revised version of the manuscript.**

Intriguingly, CBP IDRs have been described to promote the stability of the transcriptional complex by mediating multiple TFs and co-factors interactions¹⁵⁻¹⁹, supporting CBP's scaffold role in integrating combinatorial transcription factors and various transcriptional regulators to boost active transcription¹⁴. These findings expand our previous *in vitro* reporter transactivation assays, in which we tested the transactivation capacity of different CBP constructs lacking different domains. Notably, the lack of the N-terminal domain of CBP, which contains the IDR interacting with Zelda predicted by our analysis, a TAZ zinc finger domain, and a KIX domain resulted in a marked 94% decrease in CBP transactivation

power, strengthening our hypothesis that CBP function in the embryo might be related with the stabilization of TFs rather than its HAT activity⁶. Nevertheless, we cannot exclude that the disruption of such a portion could also result in unstable proteins, which could be the cause of reduced transactivation. Future experiments will shed light on the possible role of CBP as a stabilizer of TFs binding on chromatin. Taken together, we hypothesize that CBP can mediate the stabilization of other transcription factors on chromatin and promote successful zygotic genome activation.

AlphaFold3 predicted structures and interaction

References

1. Calderon, D. *et al.* The continuum of *Drosophila* embryonic development at single-cell resolution. *Science* **377**, eabn5800 (2022).
2. Ibarra-Morales, D. *et al.* Histone variant H2A.Z regulates zygotic genome activation. *Nat Commun* **12**, 7002 (2021).
3. Blythe, S. A. & Wieschaus, E. F. Zygotic genome activation triggers the DNA replication checkpoint at the midblastula transition. *Cell* **160**, 1169–1181 (2015).
4. Altendorfer, E., Mochalova, Y. & Mayer, A. BRD4: a general regulator of transcription elongation. *Transcription* **13**, 70–81 (2022).
5. Akimaru, H., Hou, D.-X. & Ishii, S. *Drosophila* CBP is required for dorsal–dependent twist gene expression. *Nat Genet* **17**, 211–214 (1997).
6. Ciabrelli, F. *et al.* CBP and Gcn5 drive zygotic genome activation independently of their catalytic activity. *Sci Adv* **9**, eadf2687 (2023).
7. Zenk, F. *et al.* Germ line–inherited H3K27me3 restricts enhancer function during maternal-to-zygotic transition. *Science* **357**, 212–216 (2017).
8. Zou, Z., Wang, Q., Wu, X., Schultz, R. M. & Xie, W. Kick-starting the zygotic genome: licensors, specifiers, and beyond. *EMBO Rep* **25**, 4113–4130 (2024).
9. Chan, S. H. *et al.* Brd4 and P300 Confer Transcriptional Competency during Zygotic Genome Activation. *Dev Cell* **49**, 867–881.e8 (2019).
10. Wang, M., Chen, Z. & Zhang, Y. CBP/p300 and HDAC activities regulate H3K27 acetylation dynamics and zygotic genome activation in mouse preimplantation embryos. *EMBO J* **41**, (2022).
11. Fleck, J. S. *et al.* Inferring and perturbing cell fate regulomes in human brain organoids. *Nature* **621**, 365–372 (2023).

12. Francois, V., Solloway, M., O'Neill, J. W., Emery, J. & Bier, E. Dorsal-ventral patterning of the *Drosophila* embryo depends on a putative negative growth factor encoded by the short gastrulation gene. *Genes Dev* **8**, 2602–2616 (1994).
13. Decotto, E. & Ferguson, E. L. A positive role for Short gastrulation in modulating BMP signaling during dorsoventral patterning in the *Drosophila* embryo. *Development* **128**, 3831–3841 (2001).
14. Ferrie, J. J. *et al.* p300 is an obligate integrator of combinatorial transcription factor inputs. *Mol Cell* **84**, 234–243.e4 (2024).
15. Chrivia, J. C. *et al.* Phosphorylated CREB binds specifically to the nuclear protein CBP. *Nature* **365**, 855–859 (1993).
16. Kwok, R. P. S. *et al.* Nuclear protein CBP is a coactivator for the transcription factor CREB. *Nature* **370**, 223–226 (1994).
17. Vo, N. & Goodman, R. H. CREB-binding Protein and p300 in Transcriptional Regulation. *Journal of Biological Chemistry* **276**, 13505–13508 (2001).
18. Nakajima, T. *et al.* RNA helicase A mediates association of CBP with RNA polymerase II. *Cell* **90**, 1107–1112 (1997).
19. Ma, L. *et al.* Co-condensation between transcription factor and coactivator p300 modulates transcriptional bursting kinetics. *Mol Cell* **81**, 1682–1697.e7 (2021).
20. Miao, L. *et al.* The landscape of pioneer factor activity reveals the mechanisms of chromatin reprogramming and genome activation. *Mol Cell* **82**, 986–1002.e9 (2022).
21. Gaskill, M. M., Gibson, T. J., Larson, E. D. & Harrison, M. M. GAF is essential for zygotic genome activation and chromatin accessibility in the early *Drosophila* embryo. *Elife* **10**, e66668 (2021).
22. Harrison, M. M., Li, X. Y., Kaplan, T., Botchan, M. R. & Eisen, M. B. Zelda Binding in the Early *Drosophila melanogaster* Embryo Marks Regions Subsequently Activated at the Maternal-to-Zygotic Transition. *PLoS Genet* **7**, e1002266 (2011).
23. Narita, T. *et al.* Enhancers are activated by p300/CBP activity-dependent PIC assembly, RNAPII recruitment, and pause release. *Mol Cell* **81**, 2166–2182.e6 (2021).
24. Flensburg, C., Kinkel, S. A., Keniry, A., Blewitt, M. E. & Oshlack, A. A comparison of control samples for ChIP-seq of histone modifications. *Front Genet* **5**, 109913 (2014).
25. Hunt, G. *et al.* Tissue-specific RNA Polymerase II promoter-proximal pause release and burst kinetics in a *Drosophila* embryonic patterning network. *Genome Biol* **25**, 1–37 (2024).
26. Liu, X. *et al.* Distinct features of H3K4me3 and H3K27me3 chromatin domains in pre-implantation embryos. *Nature* **537**, 558–562 (2016).
27. Zheng, H. *et al.* Resetting Epigenetic Memory by Reprogramming of Histone Modifications in Mammals. *Mol Cell* **63**, 1066–1079 (2016).
28. Vastenhouw, N. L. *et al.* Chromatin signature of embryonic pluripotency is established during genome activation. *Nature* **464**, 922–926 (2010).
29. Lindeman, L. C. *et al.* Prepatterning of Developmental Gene Expression by Modified Histones before Zygotic Genome Activation. *Dev Cell* **21**, 993–1004 (2011).
30. Blanco, E., González-Ramírez, M., Alcaine-Colet, A., Aranda, S. & Di Croce, L. The Bivalent Genome: Characterization, Structure, and Regulation. *Trends in Genetics* **36**, 118–131 (2020).
31. Zenk, F. *et al.* Single-cell epigenomic reconstruction of developmental trajectories from pluripotency in human neural organoid systems. *Nat Neurosci* **27**, 1376–1386 (2024).
32. Kojima, M. L., Hoppe, C. & Giraldez, A. J. The maternal-to-zygotic transition: reprogramming of the cytoplasm and nucleus. *Nat Rev Genet* **1–23** (2024).
33. Bonn, S. *et al.* Tissue-specific analysis of chromatin state identifies temporal signatures of enhancer activity during embryonic development. *Nat Genet* **44**, 148–156 (2012).
34. Koenecke, N., Johnston, J., He, Q., Meier, S. & Zeitlinger, J. *Drosophila* poised enhancers are generated during tissue patterning with the help of repression. *Genome Res* **27**, 64–74 (2017).
35. Haines, J. E. & Eisen, M. B. Patterns of chromatin accessibility along the anterior-posterior axis in the early *Drosophila* embryo. *PLoS Genet* **14**, e1007367 (2018).
36. Bozek, M. *et al.* ATAC-seq reveals regional differences in enhancer accessibility during the establishment of spatial coordinates in the *Drosophila* blastoderm. *Genome Res* **29**, 771–783 (2019).

37. Pasini, D. *et al.* Characterization of an antagonistic switch between histone H3 lysine 27 methylation and acetylation in the transcriptional regulation of Polycomb group target genes. *Nucleic Acids Res* **38**, 4958–4969 (2010).
38. Coleman, R. T. & Struhl, G. Causal role for inheritance of H3K27me3 in maintaining the OFF state of a *Drosophila* HOX gene. *Science* **356**, (2017).
39. DeLuca, S. Z., Ghildiyal, M., Pang, L.-Y. & Spradling, A. C. Differentiating *Drosophila* female germ cells initiate Polycomb silencing by regulating PRC2-interacting proteins. *Elife* **9**, 1–33 (2020).
40. Cho, C.-Y. & O’Farrell, P. H. Stepwise modifications of transcriptional hubs link pioneer factor activity to a burst of transcription. *Nat Commun* **14**, 4848 (2023).
41. Ciabrelli, F., Atinbayeva, N., Pane, A. & Iovino, N. Epigenetic inheritance and gene expression regulation in early *Drosophila* embryos. *EMBO Rep* **25**, 4131–4152 (2024).
42. Li, X.-Y., Harrison, M. M., Villalta, J. E., Kaplan, T. & Eisen, M. B. Establishment of regions of genomic activity during the *Drosophila* maternal to zygotic transition. *Elife* **3**, (2014).
43. Setty, M. *et al.* Characterization of cell fate probabilities in single-cell data with Palantir. *Nat Biotechnol* **37**, 451–460 (2019).
44. Teschendorff, A. E. & Enver, T. Single-cell entropy for accurate estimation of differentiation potency from a cell’s transcriptome. *Nat Commun* **8**, 15599 (2017).
45. Lee, T. I. *et al.* Control of Developmental Regulators by Polycomb in Human Embryonic Stem Cells. *Cell* **125**, 301–313 (2006).
46. Teschendorff, A. E., Sollich, P. & Kuehn, R. Signalling entropy: A novel network-theoretical framework for systems analysis and interpretation of functional omic data. *Methods* **67**, 282–293 (2014).
47. Rivera, J., Keränen, S. V. E., Gallo, S. M. & Halfon, M. S. REDfly: the transcriptional regulatory element database for *Drosophila*. *Nucleic Acids Res* **47**, D828–D834 (2019).
48. Oudelaar, A. M. & Higgs, D. R. The relationship between genome structure and function. *Nat Rev Genet* **22**, 154–168 (2021).
49. Zinzen, R. P., Girardot, C., Gagneur, J., Braun, M. & Furlong, E. E. M. Combinatorial binding predicts spatio-temporal cis-regulatory activity. *Nature* **462**, 65–70 (2009).
50. Mikhaylichenko, O. *et al.* The degree of enhancer or promoter activity is reflected by the levels and directionality of eRNA transcription. *Genes Dev* **32**, 42–57 (2018).
51. Kvon, E. Z. *et al.* Genome-scale functional characterization of *Drosophila* developmental enhancers in vivo. *Nature* **512**, 91–95 (2014).
52. Tie, F., Banerjee, R., Conrad, P. A., Scacheri, P. C. & Harte, P. J. Histone Demethylase UTX and Chromatin Remodeler BRM Bind Directly to CBP and Modulate Acetylation of Histone H3 Lysine 27. *Mol Cell Biol* **32**, 2323–2334 (2012).
53. Abramson, J. *et al.* Accurate structure prediction of biomolecular interactions with AlphaFold 3. *Nature* **630**, 493–500 (2024).

REVIEWERS' COMMENTS

Reviewer #1 (Remarks to the Author):

The revised manuscript has added more mechanistic insights into how CBP functions in ZGA and well addressed all my comments. Congratulations!

We sincerely appreciate the Reviewer's insightful comments and are pleased that our additional experiments and analyses have addressed the previous concerns. Overall, we believe that the Reviewer's feedback has greatly enhanced the quality and novelty of the manuscript.

Reviewer #2 (Remarks to the Author):

We have read the revised version of Cardamone et al. We appreciate the additional data and discussion which have alleviated some of our concerns. We support publication of this work if the remaining concerns - listed below - can be addressed.

Comment

1. The notion of cell-type specific chromatin landscapes in the early embryo is still a bit confusing to this reviewer. What is an early embryo? If before ZGA, then this is not shown. If during ZGA, this is kind of as expected as it is in accordance with the different lineages that can be identified by RNA-seq? While the authors have changed the wording related to this in lines 115-119, there is still the suggestion that prezygotic modifications are important for lineage specification (below) although there is no data for this modification being present in a cell-type specific manner prior to ZGA. This suggestion should be removed if no data is provided to support it.

We appreciate the Reviewer's continued engagement with our manuscript and the opportunity to clarify this important point.

We respectfully disagree with the Reviewer's assertion that we have not provided data supporting the role of pre-zygotic H3K27me3 in cell lineage specification. Our results show that H3K27me3 is already present before ZGA, but at this stage, all cells are identical, and germ layers are not yet defined. Thus, it is not possible to associate this modification with specific cell types prior to ZGA. However, **as shown in Figure 2i** and Extended Data Figure 7e, the reshaping of chromatin landscapes at ZGA occurs in a cell-type-specific manner, and **our data demonstrate that pre-zygotic H3K27me3 is essential for this process.**

We would also like to emphasize that the cell-type-specific chromatin landscape observed at ZGA is not necessarily an expected phenomenon, as **no prior studies have clearly demonstrated this.** Our findings provide strong evidence that disrupting pre-zygotic H3K27me3 or H3K27ac leads to a dramatic loss of cellular identity at the onset of ZGA, which further supports its crucial role in lineage specification.

Given this, we believe our conclusions are well-supported by the data and literature and kindly maintain our position on this matter.

Abstract

We found that the pre-zygotic modification H3K27me3 suppresses ectopic activation of marker genes across embryonic tissues by modulating cis regulatory elements, thereby ensuring the precise gene regulatory network in each germ layer during zygotic genome activation (ZGA).

Discussion

We found that histone modifications on chromatin precede the active transcription of crucial genes involved in cell fate determination.

We respectfully maintain that our findings on **pre-zygotic H3K27me3** are a key conclusion of this study, strongly supported by multiple lines of experimental evidence.

As demonstrated in Zenk et al., 2017 (Science), H3K27me3 is intergenerationally transmitted from the oocyte to the next generation (F1), justifying the use of “pre-zygotic” to describe this modification. Moreover, our **single-cell Multiome analysis, gene regulatory network modeling, and enhancer reporter assays** provide a comprehensive view of how H3K27me3 safeguards the **correct gene regulatory network at ZGA by modulating cis-regulatory elements**, ensuring lineage fidelity by **preventing the ectopic activation of marker genes**.

Additionally, we have **strong evidence** supporting the enrichment of both **H3K27me3 and H3K27ac before cell fate establishment and transcription**, which we have demonstrated in several figures of our manuscript:

- **Figure 1b** & Extended Data Figure 1b: Immunofluorescence staining for H3K27me3 and H3K27ac shows the presence of both histone marks in the nucleus **before cycle 14 (ZGA)**. These findings are further supported by our previously published studies (Zenk et al., 2017; Ciabrelli et al., 2023).
- **Figure 1c, d** & Extended Data Figure 1d, e, h: CUT&Tag analysis confirms that **H3K27me3 and H3K27ac are already enriched on chromatin before ZGA**, further reinforcing their pre-zygotic role.
- **Figure 5a** & Extended Data Figure 8b, c: CUT&Tag profiling reveals that H3K27ac (an active transcription mark) is enriched at both enhancers and promoters of genes actively transcribed at ZGA. Notably, this mark is already present **before these genes become active**, as early as cycle 9, when embryonic transcription is still quiescent.
- **Extended Data Figure 9b**: Gene Ontology (GO) analysis shows that **genes pre-marked by H3K27ac** are involved in cell fate-related processes, reinforcing the idea that these marks contribute to transcriptional priming during early development.

Given this extensive experimental support, we respectfully disagree with the Reviewer’s suggestion to remove these sentences. Our data **conclusively demonstrate** that pre-zygotic histone modifications, particularly **H3K27me3 and H3K27ac**, play an essential role in shaping the chromatin landscape prior to transcription and lineage specification.

2. The notion that CBP can stabilize pioneer factors is interesting but the data supporting it is not super strong. The loss of pioneer factors in CBP depleted embryos can be indirect, and alpha-fold is an interesting tool but of course merely a prediction (as pointed out by the authors). So the authors can suggest but not conclude that CPB stabilizes pioneer factors. Please change wording in abstract (lines 42-44), results (lines 471-474), Figure 5 caption, and discussion.

We thank the Reviewer for pointing this out. As suggested, we have changed the wording in the abstract, results, figure 5 caption, and discussion and highlighted it in orange.

Related, can the authors discuss how they think the role of CBP in stabilizing pioneer factors relates to its effect on pause release?

We thank the Reviewer for the insightful suggestion. CBP depletion and the consequent destabilization of pioneer factors might play a role in regulating RNAPII dynamics, possibly through the bromodomain protein BRD4 (Extended Data Figure 10), together with the assembly of the elongation complex. Interestingly, we could detect a mild decrease of RNAPII-S5P at Zelda-dependent genes, suggesting that CBP destabilization of this pioneer factor might impair the recruitment process as well. Indeed, Zelda RNAi embryos display decreased loading of RNAPII genome-wide, as previously described.

Reviewer #3 (Remarks to the Author):

We thank the Reviewer for the insightful comments and for significantly improving the manuscript.

Reviewer #4 (Remarks to the Author):

The authors have addressed all previous comments and provided well-analyzed new data. I have two additional suggestions regarding CBP/ZELDA interactions (no need to reply to this reviewer):

We are grateful to the Reviewer for the insightful comments, which improved the manuscript significantly.

Possible interaction between TAZ/KIX and Zelda. I suspect that these two regions might be more important than the author claimed potential coiled region. Are there any predictions from AF3 regarding these interactions?

We are grateful for the Reviewer's suggestion. The KIX domain of CBP and its role in dimerizing with transcription factors could indeed contribute to interactions with TFs. However, there is currently no evidence supporting a direct interaction between the KIX domain and Zelda. Instead, we focused on intrinsically disordered regions (IDRs) as potential mediators of this process, given that both CBP and Zelda are enriched in transcriptional condensates, which are often associated with IDR interactions. To acknowledge this possibility, we have added a statement in the discussion (lines 523–524) suggesting that more structured domains might also

contribute to protein-protein interactions. We look forward to further exploring this intriguing aspect in future studies.

For future studies, it would be interesting to perform domain tiling analysis of CBP to identify specific sequences that interact with Zelda and other TFs. While this analysis is beyond the scope of the current paper, it could provide valuable insights.

We are grateful for the Reviewer's suggestion, and we will consider this aspect for future research.